# Antibacterial and antioxidant activities of plants consumed by western lowland gorilla (*Gorilla gorilla gorilla*) in Gabon

Leresche Even Doneilly Oyaba Yinda[1]*, Richard Onanga[1], Cédric Sima Obiang[2], Herman Begouabe[2], Etienne François Akomo-Okoue[3], Judicaël Obame-Nkoghe[4], Roland Mitola[5], Joseph-Privat Ondo[2], Guy-Roger Ndong Atome[2], Louis-Clément Obame Engonga[2], Ibrahim[5], Joanna M. Setchell[6], Sylvain Godreuil[7]

1 Laboratory of Bacteriology, Interdisciplinary Medical Research Center of Franceville, Franceville, Gabon, 2 Laboratory of Research in Biochemistry (LAREBIO), University of Sciences and Technology of Masuku (USTM), Franceville, Gabon, 3 Microbiology Laboratory, Research Institute for Tropical Ecology, Libreville, Gabon, 4 Unity of Vector Ecology, Interdisciplinary Medical Research Center of Franceville, Franceville, Gabon, 5 Laboratory of Biology, University of Science and Technology of Masuku, Franceville, Gabon, 6 Department of Anthropology, Université de Durham, Durham, United Kingdom, 7 Laboratoire de Bactériologie, CHU de Montpellier, UMR MIVEGEC (IRD, CNRS, Université de Montpelier), Montpellier, France

* oyabaeven@gmail.com

**Data Availability Statement:** All relevant data can be found in the paper and its Supporting Information files.

## Abstract

Zoopharmacognosy is the study of the self-medication behaviors of non-human animals that use plant, animal or soil items as remedies. Recent studies have shown that some of the plants employed by animals may also be used for the same therapeutic purposes in humans. The aim of this study was to determine the antioxidant and antibacterial activity of *Ceiba pentandra*, *Myrianthus arboreus*, *Ficus* subspecies (ssp.) and *Milicia excelsa* bark crude extracts (BCE), plants consumed by western lowland gorillas (*Gorilla gorilla gorilla*) in Moukalaba-Doudou National Park (MDNP) and used in traditional medicine, and then to characterize their phytochemical compounds. DPPH (2,2-Diphenyl-1-Picrylhydrazyl), phosphomolybdenum complex and β-carotene bleaching methods were used to assess antioxidant activity. Antimicrobial susceptibility testing was performed using the diffusion method, while minimum inhibitory concentration (MIC) and minimum bactericidal concentration (MBC) were assessed using the microdilution method. The highest level of total phenolics was found in *Myrianthus arboreus* aqueous extract [385.83 ± 3.99 mg [gallic acid equivalent (GAE)/g]. Total flavonoid (134.46 ± 3.39) mg quercetin equivalent (QE)/100 g of extract] were highest in *Milicia excelsa*, tannin [(272.44 ± 3.39) mg tannic acid equivalent (TAE)/100 g of extract] in *Myrianthus arboreus* and proanthocyanidin [(404.33 ± 3.39) mg apple procyanidins equivalent (APE)/100 g of extract] in *Ceiba pentandra*. *Ficus* ssp. (IC$_{50}$ 1.34 ±3.36 µg/mL; AAI 18.57 ± 0.203) ethanolic BCE and *Milicia excelsa* (IC$_{50}$ 2.07 ± 3.37 µg/mL; AAI 12.03 ± 0.711) showed the strongest antioxidant activity. *Myrianthus arboreus* ethanolic BCE (73.25 ± 5.29) and *Milicia excelsa* aqueous BCE (38.67 ± 0.27) showed the strongest percentage of total antioxidant capacity (TAC). *Ceiba pentandra* ethanolic BCE (152.06 ± 19.11 mg AAE/g) and *Ficus* ssp aqueous BCE (124.33 ± 39.05 mg AAE/g) showed strongest relative antioxidant activity (RAA). The plant BCE showed antimicrobial activity against

**Funding:** The authors received no specific funding for this work.

**Competing interests:** The authors have declared that no competing interests exist.

multidrug resistant (MDR) *E. coli* (DECs) isolates, with MICs varying from 1.56 to 50 mg/mL and inhibition diameters ranging from 7.34 ± 0.57 to 13.67 ± 0.57mm. Several families of compounds were found, including total phenolic compounds, flavonoids, tannins and proanthocyanidins were found in the plant BCEs. The plant BCEs showed antioxidant activities with free radical scavenging and antimicrobial activities against 10 MDR *E. coli* (DECs) isolates, and could be a promising novel source for new drug discovery.

## 1. Introduction

Natural medicines from plants have been used to enhance human and veterinary health since time immemorial, as revealed in ancient tales, scriptures and other historical literature [1]. This practice is experiencing a resurgence [1, 2]. The World Health Organization (WHO) estimates that approximately 80% of the world's population uses medicinal plants (MPs) for their health and care needs [3]. Recent studies have focused on the potential to develop antioxidant and antimicrobial drugs from plants [4–6]. These antioxidants, for example, reduce the incidence of many metabolic diseases [7]. Antioxidants also reduce the incidence of chronic inflammation by reinforcing immunity [8], which would ultimately contribute to the efficacy of antimicrobial therapy. Antimicrobial agents are also used as antibiotics to control infections in the human body, but can cause many side effects, especially by increasing reactive oxygen species (ROS) [9]. ROS are very dangerous to human health and well-being and can contribute to the development of cancer [10]; further, they are aggravating factors for the emergence of various other metabolic diseases [11]. Finally, the use of these antimicrobial agents as antibiotic drugs can lead to resistance selection pressure in microorganisms such as bacteria [12].

Antimicrobial resistance is considered by the WHO to be one of the world's three greatest threats to human health because of the extensive spread of antibiotic-resistant bacteria (ARB) and antibiotic resistance genes (ARGs) [13]. Infectious diseases caused by multidrug-resistant (MDR) bacteria affect millions of people worldwide [14]. Furthermore, many zoonoses caused by pathogenic microorganisms such as enterobacteria (e.g.; MDR *Escherichia coli*), have been a human public health problem for decades [15]. *E. coli* is a Gram-negative bacteria and gut commensal in animals, including non-human primates (NHPs) [16]. The close phylogenetic relationship between humans and other hominids, combined with a rapidly expanding human–animal interface, enables pathogen transmission across species, leading to morbidity and mortality in great ape populations throughout the world [17, 18]. Several studies have shown that wildlife [19, 20], including wild primates such as western lowland gorillas (*Gorilla gorilla gorilla*), could transmit this kind of pathogens to humans [21, 22], or that humans could transmit the pathogens to wildlife [23, 24]. This situation has great potential for the discovery of new antimicrobial agents [25, 26]. Many recent studies show that natural products from medicinal plants or plants consumed by animals continue to play a role in drug discovery and development [27, 28].

Zoopharmacognosy is the study of non-human animals self-medicating, using plants, animals and natural compounds, such as soil, as a preventative or direct medicinal cure to regain health in their natural habitat [2, 29]. Most great apes, including western lowland gorillas (*Gorilla gorilla gorilla*) have a predominantly frugivorous diet [30, 31]. However, bark is the main fallback food for gorillas [32]. These feeding practices appear to be beneficial to the well-being and health of these animals but also to those of humans [33]. The use of great ape pharmacopoeia or zoopharmacognosy is a very promising strategy for management of human

diseases because of the phylogenetic proximity of humans and great apes [34]. Several studies have shown that plants from the diet of great apes, including western lowland gorilla, are also used as MPs by healers in traditional African medicine [35, 36].

In view of the physiological (pathological, infectious) state of the gorillas in the PNMD, linked to the presence of potentially pathogenic enterobacteria, multi-resistant to antimicrobials used in human therapy, including MDR enterobacteria such as MDR *E. coli* (DECs) obtained in a previous study, how did these animals manage to host and control these microorganisms? In this study, we hypothesized that "the immunity-enhancing consumption of certain plant items (such as bark) by gorillas could be responsible for their ability to host and control these infectious microorganisms without developing serious disease".

Gabon, with its exceptional biodiversity, constitutes a vast reservoir of unexplored potential active biomolecules [37]. This study aims to evaluate the chemical composition (secondary metabolites families), antioxidant and antimicrobial activities of four plant species consumed by gorillas living in MDNP to control microbial infections within their communities and, used as traditional MPs by healers in Gabon. *Ceiba pentandra* [38, 39], *Myrianthus arboreus* [40, 41], *Ficus* ssp [42] and *Milicia excelsa* [43, 44] are the four plants selected for this study on the basis of ethnobotanical and ethnopharmacological surveys carried out among local populations. The ethnopharmacological activities of these four plants in traditional medicine have already been reported in recent literature.

## 2. Material and methods

### 2.1. Study area and field research authorization

Sampling of the bark of four plants consumed by western lowland gorillas was carried out under field research authorization N° 003/20/DG/JBLD/N° 306, from August 1 to 11, 2022 in MDNP, during the daily monitoring of gorillas by observing plants items they consumed in their natural environment during this period, using the non-invasive method as previously described [45]. Research Institute for Tropical Ecology (IRET) provided all the necessary authorizations to carry out this study on the MDNP site. No special permits were required, due to the scientific and technical agreements established between the Centre National de Recherches Scientifiques et Techniques (CENAREST) and the Centre Interdisciplinaire de Recherches Médicales de Franceville (CIRMF).

### 2.2. Data and sample collection

A survey was carried out among 27 inhabitants of the village of Doussala, including a number of traditional healers and herbalists, both men and women, known to the local peoples, using a questionnaire, as previously described by Obiang et al. [46]. Fresh bark plant samples collected were identified by a team of botanists leading by Prof. Brama Ibrahim from the Department of Biology, Faculty of Sciences of the University of Science and Technology of Masuku (USTM) in Franceville (Gabon). Voucher specimens of *Ceiba pentandra* (BRLU/4618), *Myrianthus arboreus* (BRLU-LBV/8324), *Ficus* ssp (USTM/#443) and *Milicia excelsa* (USTM/739) has been deposited at the herbarium of the same institute. The choice of the bark of the selected plants was made based on local traditional medicinal use and the fact that gorillas also consumed them.

**2.2.1. Therapeutic indications.** Descriptive statistical methods were used to analyze the ethnopharmacological survey data and various quantitative indices, including use value (UV) and relative frequency of citation (RFC). Data were reported in proportions and percentages [46].

Use Value (UV) and Relative Frequency of Citation (RFC) were calculated according to the following formula:

$$UV = U/n \text{ and } RFC = FC/N (0 < RFC < 1).$$

## 2.3. Treatment of plant material

**2.3.1. Extraction.** From 100 g of bark powder of each selected plant, an extraction by maceration under agitation was carried out for 72 h with 2 L of each solvent (ethanol 99.8% and water). After filtration of the two (ethanolic and aqueous) maceration, using Whatman N°1 filter paper, the aqueous extract was directly lyophilized, while the ethanolic extract was concentrated and dried in an oven.

## 2.4. Preliminary phytochemical screening

Each plant BCE was tested for the presence of flavonoids, coumarins, tannins, total phenolics, catechin tannins, gallic tannins, cyanidins, alkaloids, oses or holosides, saponosides, sterols or triterpenoids, anthracenosides, cardiotonic heterosides and reducing sugar as described previously [47, 48].

## 2.5. Quantitative phytochemical analysis

**2.5.1. Total phenol content.** To determine the total phenol content, the Folin-Ciocalteu method was used [49]. Absorbance was measured at 735 nm. Phenolic compounds were expressed as mg gallic acid equivalents (GAE) /dry weight of extract.

**2.5.2. Total flavonoid content.** Aluminum trichloride method was used to determine the flavonoid content and absorbance was measured at 435 nm. Flavonoid content was expressed in quercetin equivalent (QE) [50].

**2.5.3. Tannin content.** Tannin content was determined using the method described previously by Sima-Obiang et al. [51]. Absorbance was measured at 525 nm and tannic acid was used as a standard. Tannin contents were expressed in mg of tannic acid equivalent (TAE)/g of dry extract.

**2.5.4. Proanthocyanidin content.** Proanthocyanidins were determined using the HCl-Butanol method [52]. Absorbance was read at 550 nm and apple procyanidin was applied as standard. Proanthocyanidin levels were expressed in apple procyanidins equivalent (APE)/g of dry extract.

## 2.6. Bioactive properties of bark crude extracts of selected species consumed by western lowland gorilla

**2.6.1. Antioxidant activity of bark crude extracts of selected species consumed by western lowland gorilla.** *2.6.1.1. 2,2-Diphenyl-1-Picrylhydrazyl (DPPH) Radical Capacity.* The method described by Scherer and Godoy [53], based on the DPPH (2, 2-diphenyl-1-picrylhydrazyl) radical test, was used to determine the Antioxidant Activity Index (AAI). Briefly, DPPH solution was prepared by dissolving 10 mg of DPPH powder in 200 mL methanol ([DPPH] = 0.05 mg/mL). 400 μL of each of the eight BCE (at 1mg/mL concentration) were added to 1.6 mL of methanol to obtain a stock solution. Cascade dilutions to 1/2 were made from the stock solution into test tubes containing 1 mL of methanol beforehand. Then, 1 mL of DPPH was added to each of the tubes. Absorbencies were measured at 517 nm after 30 min incubation at room temperature in the dark against a blank. Ascorbic acid (vitamin C) was used as reference. The ability to scavenge DPPH radical (RSA) was calculated by the following

equation:

$$\% \text{ RSA(Relative Scavenging Activity)} = [(A_{\text{control}} - A_{\text{sample}})/A_{\text{control}}] \times 100.$$

A = Absorbance at 517 nm. The $IC_{50}$ (concentration providing 50% inhibition) of BCE and standards was determinate using regression curves in the linear range of concentrations. The AAI was then calculated as follows:

$$\text{AAI(Antioxidant Activity Index)} = [\text{DPPH] f}(\mu\text{g.mL}^{-1})/\text{IC50}(\mu\text{g.mL}^{-1}).$$

[DPPH] f is the final concentration of DPPH.

*2.6.1.2. Phosphomolybdenum complex method for total antioxidant capacity.* Spectrophotometric evaluation of total antioxidant activity was carried out through the formation of a phosphomolybdenum complex. The assay was based on the reduction of Mo (VI) to Mo (V) and subsequent formation of a green phosphate/Mo (V) complex in acid pH [54, 55]. A total volume of 0.3 mL of each bark crude extract (at 5mg/mL concentration) dissolved in methanol was added to 3 mL of reagent solution (0.6 mol/L $H_2SO_4$, 28 mmol/L $Na_3PO_4$ and 4 mmol/L ammonium molybdate). The mixtures were incubated at 95°C for 90 min the cooled to room temperature. The absorbance was measured at 695 nm. The total antioxidant activity of each plant BCE was expressed as the number of equivalence of ascorbic acid (mg AAE/g).

*2.6.1.3. β-Carotene bleaching assay.* The β-carotene–linoleic acid model system is prepared by dissolving 0.5 mg β-carotene in 1 mL chloroform followed by the addition of 40 μL linoleic acid and 500 μL Tween-20. A rotary evaporator was used to completely evaporate the chloroform added into the system. To the resulting solution, 100 mL of oxygenated distilled water is added followed by vigorous shaking. 500 μL of BCE (at 1mg/mL concentration) or antioxidant solution of the reference (Ascorbic acid; 1mg/mL concentration) was added to 2.5 mL of the previous emulsion. The absorbance was measured at 490 nm before and after heat treatment with regular time intervals for 48 h. The measurement of the absorbance continued until the color of β-carotene disappears [56].

The bleaching analysis of β-carotene was calculated as follows:

$$\text{RAA(Relative Antioxidant Activity)} = [\text{Abst:}_{48h}(\text{sample})/\text{Abst:}_{48h}(\text{Vit C})] \times 100\%.$$

**2.6.2. Antimicrobial activity of bark crude extracts of selected species consumed by western lowland gorilla.** *2.6.2.1. Bacterial strains tested growth conditions and inoculums standardization.* Ten MDR *E. coli* (DECs) isolates obtained in a previous study [57] were used to assess the plant studied BCE antimicrobial activity. The cultures were held at 4°C on Muller-Hinton agar (bioMérieux, France). Colonies from 24-h cultures were used to make the inoculum suspension. The colonies were vortexed for 15 s after being suspended in sterile saline (0.9% NaCl). The turbidity of a 0.5 McFarland Norm (equivalent to 1–5 $10^8$ CFU/mL) was used as density setting [58].

*2.6.2.2. Antimicrobial activities.* To test the antimicrobial activity of BCEs and the antimicrobial susceptibility, antibiograms were performed using the Kirby–Bauer disk diffusion method, according to the CLSI (Clinical and Laboratory Standards Institute) protocols [59]. The agar was suspended in distilled water, heated to complete dissolution autoclaved at 121°C, and poured into Petri dishes. Whatman paper discs (6 mm) were inoculated with suspensions ($10^8$ CFU/mL) and dispersed on the surface of Mueller-Hinton agar plates (bioMérieux, France) [60]. Then, the discs were impregnated with 20 μL of each bark crude plant extracts. All plates were incubated for 24 h at 37°C. The diameters of the inhibition zones were

determined after incubation. Antimicrobial activity was estimated after incubation at 37˚C for 24 h by measuring the zone of inhibition against the tested microorganisms. Antimicrobials tested were amoxicillin/clavulanic acid (AMC, 30 μg) and gentamycin (GEN, 10 μg). Break-points provided by the CLSI were used for the designation of isolates as resistant (R), interme-diately susceptible (IS) or susceptible (S). For subsequent data analysis, the isolates with an I result were grouped with the isolates that gave an R result and defined as resistant. Multidrug-resistant isolates were identified based on the definition of MDR as bacteria that are resistant to three or more classes of antimicrobial agents [61, 62].

*2.6.2.3. Minimum Inhibitory Concentration (MIC) and Minimal Bactericidal Concentration (MBC) assays.* In microplate wells (96 wells), 10 μL of each dilution of plant bark crude extracts varying from 30 mg/mL to 0.014 mg/mL were combined with Mueller Hinton broth (bioMér-ieux, France) (170 μL) and bacterial inoculums (20 μL) and set to a final microbial concentra-tion of $5 \times 10^5$ CFU/mL according to NCCLS standards methods, with some modifications [63]. The ethanol content in each well was less than 3.5% in an overall amount of 200 μL. For the negative control, the same percentage of ethanol was used. The MIC is the lowest concen-tration that does not emit a red color after 2 h of incubation. To assess MBC, a portion of each well where the concentrations are > or = (MIC) was sub-cultured on Muller-Hinton agar (MHA) (bioMérieux, France) and incubated for 24 h at 37˚C. The MBC is described as the extract concentration at which 99.9% of the inoculated bacteria were destroyed [64].

## 2.7. Data analysis

All tests were performed as triplicate and the results are showed present as the mean. Excel software for Microsoft was used to analyze data. Analysis of variance (ANOVA) followed by Student's tests were used to test for significant differences between means. Differences were considered statistically significant at $p < 0.05$. Adobe Illustrator software was used to plot the histograms. The Pearson correlation was determined between the antimicrobial and antioxi-dant activity of plant BCE and total phenolic, flavonoid, proanthocyanidin and tannin content. To visualize the correlation data, a heatmap of three colors: red (r = ―1), white (r = 0) and blue (r = -1) analysis was plotted with the packages Gplot using R 4.0.2.

## 2.8. Inclusivity in global research

Additional information regarding the ethical, cultural, and scientific considerations specific to inclusivity in global research is included in the S1 Checklist.

## 3. Results

### 3.1. Ethnobotanical and ethnopharmacological survey

Western lowland gorilla living in MDNP consumed 27 plants [30]. Local people use different parts of these plants (bark, root, fruit and leaves) in medicinal preparation (maceration, decoc-tion, lotion, pomade and infusion) (Table 1). The cross-referenced results of ethnobotanical and ethnopharmacological surveys on the traditional use of plants by traditional healers in their pharmacopoeia to treat various human illnesses enabled us to select the four plants con-sumed by western lowland gorillas living in MDNP for this study. This information was recov-ered from autochthone Vungu people, living in Doussala village in MDNP.

Table 2 shows the results of ethnobotanical and ethnopharmacological surveys about tradi-tional uses of most cited *Ceiba pentandra* L., *Myrianthus arboreus* P. Beauv., *Ficus ssp*. and *Milicia excelsa* Welw C.C. berg. bark by traditional healers in their pharmacopoeia to treat var-ious human diseases, plants consumed by western lowland gorillas living in MDNP. These

**Table 1. Plant species consumed by gorillas and used by humans to treat various diseases including diarrhea in southeast Gabon.**

| Species | Family | Local name in Vungu | Part used | Preparations | Route of administration | Indication | UV | RFC | Nature |
|---|---|---|---|---|---|---|---|---|---|
| *Caloncoba welwitschii* | Flacourtiaceae | myanmongom | Leave, root, bark, fruit, seed | Decoction, macération, infusion | Oral administration | asthma, heavy, painful periods, harmless madness, biliary disorders | 0,19 | 0,02 | Tree |
| *Cissus dinklagei* | Verbenaceae | eéko / léko | Fruit, liana, leave | Raw, dried, cooked | Oral administration, ocular route | myopia, yellow fever, diarrhoea, stomach pain | 0,12 | 0,02 | Tree |
| *Ceiba pentandra.* | Malvaceae | Mufuma | Fruit, bark, fruit, root, leave | Decoction, macération, infusion | Oral administration, baths | colic, diarrhoea, inflammation, poisoning, fatigue, diurétique, hydropisia | 0,22 | 0,0352 | Tree |
| *Cola sp.* | Sterculiaceae | fudi | Seed, fruit | Mastication | Oral administration | stimulant, appetite suppressant | 0,07 | 0,0315 | Tree |
| *Dichostema glaucescens* | Euphorbiaceae | Dibule / Mabule | Bark | maceration, powdered | Oral administration | bad luck, purification bath, nurturing galactogen, emetic, diarrhoea | 0,15 | 0,016 | Tree |
| *Diospyros mannii* | Ebenaceae | Emba | Fruit, leave, seed, bark | bark shavings, | Cutaneous application, lotions | chest pain | 0,04 | 0,026 | Tree |
| *Diospyros sp.* | Ebenaceae | mufi nzi | Fruit, leave, seed | Decoction, macération, drink | Oral administration | albumin regulation, cysticercosis, intestinal parasites, toxoplasmosis, allergies, emphysema | 0,22 | 0,02 | Tree |
| *Diospyros spp.* | Ebenaceae | nemba / ilalaba | Bark, root, stem, fruit | Decoction, macération | Oral administration | antibacterial, antifungal, diarrhoea | 0,07 | 0,02 | Tree |
| *Duboscia macrocarpa* | Tiliaceae | Moupighi | Fruit, leave, seed, bark | Decoction, macération, infusion | Oral administration | toothache, coughing, tuberculose, vermifuge, stomach pain | 0,19 | 0,026 | Tree |
| *Eriocoelum macrocarpum* | Sapindaceae | Dibotsa | Bark, root | Decoction | Oral administration | female infertility, food cooking liquid | 0,08 | 0,0178 | Tree |
| *Ficus* ssp | Moraceae | divevenguengui | Fruit, leave, seed, bark | Decoction, macération, infusion, latex | Oral administration, cutaneous application, lotions | painful periods, diarrhoea, uterine care, local application | 0,12 | 0,0352 | Tree |
| *Hexalobus crispifl orus* | Annonaceae | tsago | Leave, root, bark, fruit | Decoction, macération | Oral administration | venereal diseases, paludisme, wounds, boils, fever, muscle pain, arthritis, convulsion | 0,3 | 0,0315 | Tree |
| *Irvingia gabonensis* | Irvingiaceae | andok / Mwiba | Fruit, leave, seed, bark | Raw, dried, decoction, eaten, fresh, macération | Oral administration, cutaneous application, lotions | deconstipant, diarrhoea or dysentery, toothache, astringent | 0,15 | 0,013 | Tree |
| *Klainedoxa gabonensis* | Irvingiaceae | mougoma | Fruit, leave, seed, bark | Raw, dried, decoction, eaten, fresh | Oral administration, cutaneous application, lotions | analgesic, venereal disease, diarrhoea, sterility, impotence | 0,12 | 0,026 | Tree |
| *Lannea welwitschii* | Anacardiaceae | gongo | Fruit, leave, seed | Decoction, macération | Oral administration | AIDS, opportunistic diseases, respiratory tract infections, toothache, hypertension | 0,19 | 0,02 | Tree |

*(Continued)*

**Table 1.** (Continued)

| Species | Family | Local name in Vungu | Part used | Preparations | Route of administration | Indication | UV | RFC | Nature |
|---------|--------|---------------------|-----------|--------------|-------------------------|------------|-----|-----|--------|
| *Meiocarpidium lepidotum* | Annonaceae | depeyrie | Bark, root, fruit | Decoction, macération | Oral administration | fever, purgative, abdominal pain, worm infections in babies | 0,15 | 0,016 | Tree |
| *Milicia excelsa* | Moraceae | Kambal / Iroko | Leave, root, bark, fruit | Decoction, macération, infusion, latex | Oral administration, cutaneous application, lotions, ocular route | filariose, schizophrenia, diarrhoea, infertility, fortifiant | 0,19 | 0,0352 | Tree |
| *Myrianthus arboreus* | Moraceae | dibimbi / mububa | Fruit, leave, seed, bark | Raw, dried, decoction, macération | Oral administration | infertility, stomach ulcer, sore, gonorrhea, hypertension, cough, anemia | 0,26 | 0,0352 | Tree |
| *Panda oleosa* | Pandaceae | ovaga | Leave, root, bark, fruit, seed | Decoction, macération, infusion | Oral administration | burns, diarrhoea, dysmenorrhea, milk purification for nursing mothers, analgésique | 0,19 | 0,016 | Tree |
| *Pycnanthus angolensis* | Myristicaceae | mulomba | Fruit, leave, liana, seed, bark | Decoction, eaten, macération, drink | Oral administration, cutaneous application, lotions | stomach pain, diarrhoea, enema, madness, epilepsy, gastritis | 0,19 | 0,0315 | Tree |
| *Ricinodendron heudelotii* | Euphorbiaceae | ndjoé | Fruit, leave, seed, bark | Decoction, macération | Oral administration | AIDS, opportunistic diseases, respiratory tract infections, toothache, hypertension, stomach pain | 0,22 | 0,0178 | Tree |
| *Sacoglottis gabonensis* | Humiriaceae | Ozouga | Fruit, bark, fruit | Decoction, macération | Oral administration | AIDS, opportunistic diseases, venereal diseases | 0,12 | 0,0123 | Tree |
| *Synsepalum dulcifi cum* | Sapotaceae | | Fruit | Raw, dried, cooked | Oral administration | suppress sensations of acidity and bitterness | 0,04 | 0,026 | Tree |
| *Staudtia gabonensis* | Myristicaceae | mobé / mibé | Fruit, bark | Decoction, macération | Oral administration | haemostatic, intestinal worms, anemia, menstruation | 0,15 | 0,0178 | Tree |
| *Trichilia prieureana* | Meliaceae | | Fruit, leave, seed, bark | Decoction, macération, infusion | Oral administration, cutaneous application, lotions | venereal diseases, fever, coughs, constipation, poisoning, ascites, aphrodisiac, lumbago, rheumatism | 0,34 | 0,013 | Tree |
| *Uapaca guineensis* | Euphorbiaceae | Assam | Fruit, bark, fruit, root | Decoction, macération | Oral administration | intestinal parasitism, anti-abortive, aphrodisiac, restorative for young mothers, leprosy, gastrointestinal disorders, diarrhoea | 0,22 | 0,02 | Tree |
| *Xylopia guintasii* | Annonaceae | M'voma | Bark, root | Decoction, eaten, macération, drink, powdered | Oral administration, cutaneous application, lotions | broncho-pneumonic affections, febrile pains, knot-like swellings, treatment of pyorrhoea, ulcers | 0,19 | 0,013 | Tree |

UV: Use Value indicates the relative importance of uses of plant species. RFC: Relative Frequency of Citation indicates the local importance of each species.

informations were recovered from autochthone people, called Vungus, living in Doussala village in MDNP and the relevant literature review.

## 3.2. Extraction yields

Extraction yields varied according to the solvents used. Water proved more efficient with higher yields than ethanol.

**Table 2. Phytochemical screening of *Ceiba pentandra*, *Myrianthus arboreus*, *Ficus* ssp and *Milicia excelsa*.**

| Secondary metabolites | | *Ceiba Pentandra* | | *Myrianthus arboreus* | | *Ficus* ssp | | *Milicia excelsa* | |
|---|---|---|---|---|---|---|---|---|---|
| | | AE | EE | AE | EE | AE | EE | AE | EE |
| Saponins | | ++ | + | ++ | + | ++ | + | - | - |
| Tannin gallic | | +++ | ++ | +++ | ++ | ++ | ++ | ++ | ++ |
| Tannin catechin | | ++ | ++ | ++ | ++ | ++ | ++ | ++ | ++ |
| Total phenolics | | +++ | ++ | +++ | ++ | +++ | ++ | +++ | ++ |
| Total flavonoids | | ++ | ++ | ++ | + | - | - | ++ | ++ |
| Reducing sugars | | +++ | ++ | +++ | ++ | + | ++ | ++ | ++ |
| Alkaloids | | +++ | ++ | +++ | ++ | +++ | ++ | +++ | ++ |
| Anthracenosides | | +++ | ++ | ++ | ++ | ++ | ++ | - | - |
| Coumarins | | +++ | ++ | +++ | ++ | +++ | ++ | +++ | ++ |
| Sterols and Triterpenoids | | + | + | + | - | + | - | + | - |
| Oses and holosides | | +++ | ++ | - | + | + | ++ | ++ | ++ |
| Cardiac glycosides | | +++ | ++ | +++ | ++ | ++ | ++ | +++ | ++ |
| | Digitoxins | +++ | ++ | +++ | + | +++ | ++ | +++ | ++ |
| | Gitoxins | - | - | - | - | ++ | ++ | ++ | ++ |
| | Gitoxins genins | - | - | +++ | ++ | | | | |
| Cyanidins | | +++ | ++ | +++ | ++ | - | - | +++ | ++ |
| | Flavonols | - | - | - | - | - | - | - | - |
| | Flavones | +++ | + | ++ | + | - | - | - | - |
| | Flavonones | - | - | - | - | - | - | ++ | ++ |

+++ = very abundant; ++ = abundant; + = not abundant,— = not detected. EE: Ethanol extract; AE: Aqueous extract. Color intensity/foam observed was used to indicate the phytochemicals abundance.

### 3.3. Preliminary phytochemical screening

Phenolic compounds, alkaloids, flavonoids, tannin gallic, anthracenosides, reducing sugar, coumarins, sterols and triterpenes, oses and holosides, cyanidines, cardiac glycosides and saponins were found in almost all plant BCEs tested (Table 4).

### 3.4. Antioxidant potential of bark crude extracts of selected species consumed by western lowland gorilla

The results of antioxidant potential of the four plant bark crude extracts are shown in Figs 1 and 2.

### 3.5. Antioxidant activity of bark crude extracts of selected species consumed by western lowland gorilla

**3.5.1. DPPH radical scavenging activity.** The AAI of the bark crude extracts from *Ceiba pentandra* ranged from 9.3 ± 1.28 to 10.17 ± 3.45. For *Myrianthus arboreus* it ranged 11.81 ± 0.30 to 13.72 ± 6.88. From *Ficus* ssp it ranged 7.11 ± 1.16 to 19.45 ± 2.98, and from *Milicia excelsa* it ranged 7.11 ± 4.64 to 11.98 ± 3.56 (Table 3). These values are lower than the AAI of Ascorbic acid (23.16 ± 20.26).

**3.5.2. Phosphomolybdenum complex assay.** *Myrianthus arboreus* aqueous extract showed the highest total antioxidant activity with 152.06 ± 19.11 mg AAE/g and *Ceiba pentandra* aqueous extract showed the lowest value with 54.73 ± 12.48 mg AAE/g (Table 3).

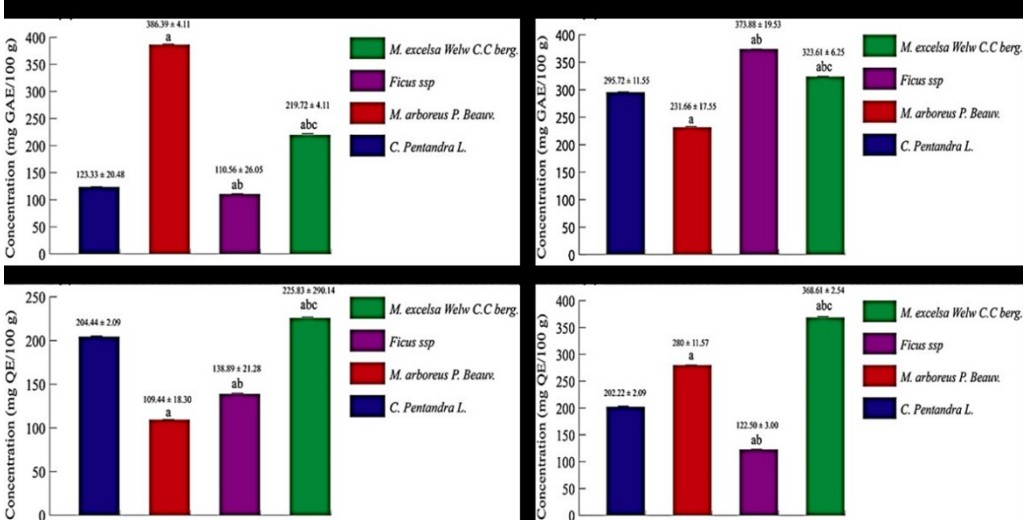

**Fig 1. Total phenolic and flavonoid content of *Ceiba pentandra*, *Myrianthus arboreus*, *Ficus* ssp and *Milicia excelsa* BCEs.** (a): Total phenolic content of plant bark crude aqueous extracts. (b): Total phenolic content of plant bark crude ethanolic extracts. (c): Total flavonoid content of plant bark crude aqueous extracts. (d): Total flavonoid content of plant bark crude ethanolic extracts. [a] $p < 0.05$ compared to *Ceiba pentandra* BCE in the same column; [b] $p < 0.05$ compared to *Myrianthus arboreus* BCE in the same column; [c] $p < 0.05$ compared to *Ficus* ssp BCE in the same column; GAE = gallic acid equivalent; QE = quercetin equivalent; TAE = tannic acid equivalent; APE = apple procyanidins equivalent.

### 3.5.3. β-Carotene bleaching assay.

The β-carotene bleaching assay results (Table 3) and β-carotene bleaching kinetics (Fig 3) showed that the highest relative antioxidant activity (RAA) was shown by *Ceiba pentandra* ethanolic extract (RAA = 73.25 ± 5.29) and *Ficus* ssp

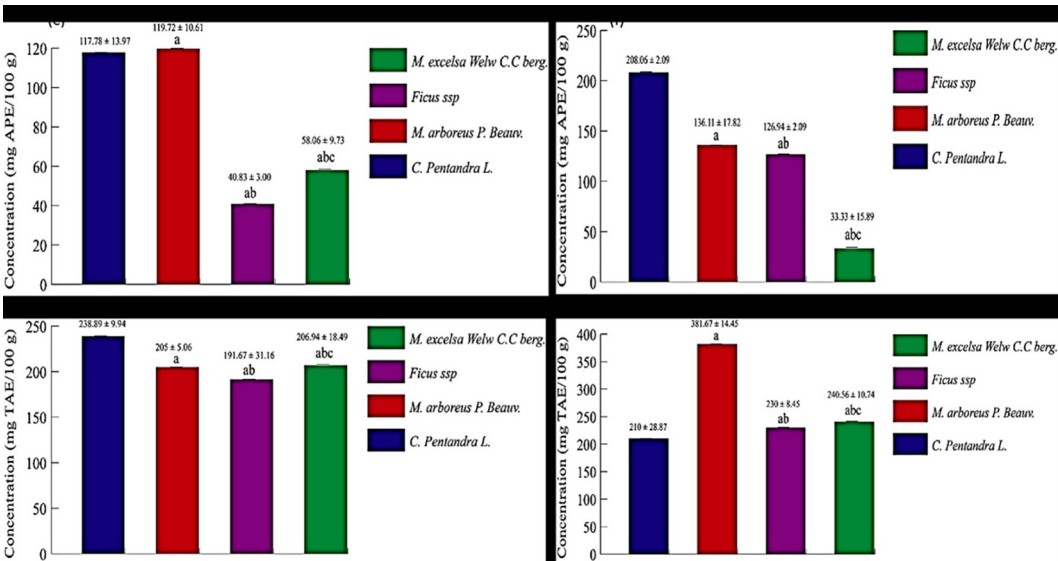

**Fig 2. Total proanthocyanidin and tannin content of *Ceiba pentandra*, *Myrianthus arboreus*, *Ficus* ssp and *Milicia excelsa* BCE.** (e): Total proanthocyanidin content of plant bark crude aqueous extracts. (f): Total proanthocyanidin content of plant bark crude ethanolic extracts. (g): Total tannin content of plant bark crude aqueous extracts. (h): Total tannin content of plant bark crude ethanolic extracts. [a] $p < 0.05$ compared to *Ceiba pentandra* BCE in the same column; [b] $p < 0.05$ compared to *Myrianthus arboreus* BCE in the same column; [c] $p < 0.05$ compared to *Ficus* ssp BCE in the same column; GAE = gallic acid equivalent; QE = quercetin equivalent; TAE = tannic acid equivalent; APE = apple procyanidins equivalent.

**Table 3. Antioxidant activities of *Ceiba pentandra*, *Myrianthus arboreus*, *Ficus* ssp and *Milicia excelsa* BCE determined by DPPH, Phosphomolybdenum complex and β-Carotene bleaching assays using L-ascorbic acid as a positive control.**

| Sample | DPPH IC50 (mg/mL) | DPPH AAI | PM TAC (mg AAE/g) | BCB RAA (%) |
|---|---|---|---|---|
| *Ceiba Pentandra* | | | | |
| AE | 2.73 ± 0.35 | 9.3 ± 1.28 | 54.73 ± 12.48 | 37.07 ± 0.44 |
| EE | 2.65 ± 0.87 | 10.17 ± 3.45 | 48.66 ± 12.96 | 73.25 ± 5.29 |
| *Myrianthus arboreus* | | | | |
| AE | 2.12 ± 0.05[a] | 11.81 ± 0.30[a] | 152.06 ± 19.11[a] | 36.47 ± 0.64[a] |
| EE | 2.18 ± 1.12[a] | 13.72 ± 6.88[a] | 104.33 ± 63.25[a] | 36.35 ± 0.44[a] |
| *Ficus* ssp | | | | |
| AE | 6.15 ± 0.24[a,b] | 7.11 ± 1.16[a,b] | 90.6 ± 49.77[a,b] | 38.67 ± 0.27[a,b] |
| EE | 1.31 ± 0.20[a,b] | 19.45 ± 2.98[a,b] | 92.93 ± 20.50[a,b] | 29.83 ± 0.20[a,b] |
| *Milicia excelsa* | | | | |
| AE | 6.15 ± 6.14[a,b,c] | 7.11 ± 4.64[a,b,c] | 124.33 ± 39.05[a,b,c] | 34.87 ± 0.35[a,b,c] |
| EE | 2.25 ± 0.79[a,b,c] | 11.98 ± 3.56[a,b,c] | 104.6 ± 36.61[a,b,c] | 31.79 ± 0.57[a,b,c] |
| Ascorbic acid | 1.73 ± 1.20[a,b,c,d] | 23.16 ± 20.26[a,b,c,d] | | 100.00 ± 1.78[a,b,c,d] |

DPPH: (2,2-Diphenyl-1-Picrylhydrazyl); $IC_{50}$: the concentration of extracts reducing 50% of DPPH; AAI: Antioxidant Activity Index; PM TAC: Phosphomolybdenum Total Antioxidant Capacity; BCB RAA: β-Carotene Bleaching Relative Antioxidant Activity. Values are expressed as means ± SD. [a] $p < 0.05$ compared to aqueous and ethanolic extracts of *Ceiba Pentandra*. in the same column; [b] $p < 0.05$ compared to aqueous and ethanolic extracts of *Myrianthus arboreus* in the same column; [c] $p < 0.05$ compared *Ficus* ssp in the same column; [d] $p < 0.05$ compared *Milicia excelsa* in the same column.

aqueous extract (RAA = 38.67 ± 0.27). *Ficus* ssp and *Myrianthus arboreus* ethanolic extract showed lower activities (RAA = 31.79 ± 0.57).

### 3.6. Antimicrobial activity of bark crude extracts of selected species consumed by western lowland gorilla

The ethanolic BCE tested showed more significant activity against MDR *E. coli* (DECs) than the aqueous BCE. Of the four plants tested, all extracts were active against at least one MDR *E. coli* (DECs) isolate (Table 4 and Fig 4). *Ceiba pentandra* aqueous BCE showed remarkable activity against all the tested MDR *E. coli* (DECs) isolates with inhibition zones ranging from 8.34 ± 0.57 mm to 13.67 ± 0.57 mm (Table 4). *Ceiba pentandra* ethanolic BCE also showed reasonable activity against all the tested MDR *E. coli* (DECs) isolates with inhibition zones ranging from 8.34 ± 0.57 mm to 11.67 ± 0.57 mm. *Ficus* ssp aqueous and ethanolic BCE showed activity against MDR *E. coli* (DECs) isolates with inhibition zones ranging from 8.34 ± 0.57 mm to 10.4 ± 0.57 mm and 7.67 ± 1.54 mm to 10.67 ± 0.57 mm respectively. *Myrianthus arboreus* aqueous BCE showed activity only against two MDR *E. coli* (DECs) isolates with inhibition zones ranging from 7.34 ± 0.57 mm to 8.34 ± 0.57 mm. *Myrianthus arboreus* ethanolic BCE showed activity against six tested MDR *E. coli* (DECs) isolates with inhibition zones ranging from 8.67 ± 0.57 mm to 10.67 ± 0.57 mm. *Milicia excelsa* Welw C.C berg. aqueous BCE were the least active. Ethanol (98%) and water were used as control and did not have any effect on all studied MDR *E. coli* (DECs) isolates.

MIC values were between 1.56 mg/mL and 50 mg/mL, and MBC values were between 3.12 mg/mL and >50 mg/mL (Table 5).

### 3.7. Correlations analysis

There was a positive correlation between the overall total phenolic, flavonoid, proanthocyanidin and tannin content and antimicrobial activity and a negative correlation with antioxidant activity assessed by DPPH $IC_{50}$, PM TAC and BCB RAA (Figs 5 and 6).

**Table 4. Inhibition zones (mm) diameters induced by *Ceiba pentandra*, *Myrianthus arboreus*, *Ficus* ssp and *Milicia excelsa* BCE against MDR *E. coli* (DECs).**

| Species | E32 | E40 | E41 | E8 | E4 | E7 | E37 | E50 | E49 | E10 |
|---|---|---|---|---|---|---|---|---|---|---|
| | | | | | MDR *E. coli* (DECs) | | | | | |
| *Ceiba Pentandra* | | | | | | | | | | |
| AE | 8.67 ± 0.57 | 9.34 ± 0.57 | 8.34 ± 0.57 | 8.67 ± 0.57 | 9.34 ± 0.57 | 13.67 ± 0.57 | 10.34 ± 0.57 | 13.34 ± 0.57 | 8.34 ± 0.57 | 9.67 ± 0.57 |
| EE | 9 ± 0.57 | 9.67 ± 0.57 | 9.67 ± 0.57 | 11.67 ± 0.57 | 10.34 ± 0.57 | 11.67 ± 0.57 | 10.34 ± 0.57 | 9.34 ± 0.57 | 8.34 ± 0.57 | 10.34 ± 0.57 |
| *Myrianthus arboreus* | | | | | | | | | | |
| AE | 7.34 ± 0.57 | 8.34 ± 0.57 | ND | ND | ND | ND | ND | ND | ND | ND |
| EE | 8.67 ± 0.57 | 9 ± 0.57 | ND | ND | ND | 10.67 ± 0.57 | 10.34 ± 0.57 | 10.34 ± 0.57 | 10 ± 0.57 | ND |
| *Ficus* ssp | | | | | | | | | | |
| AE | ND | ND | 8.67 ± 0.57 | 8 ± 0.57 | 8 ± 0.57 | 8 ± 0.57 | 9.67 ± 0.57 | 8.34 ± 0.57 | 10.4 ± 0.57 | ND |
| EE | 9 ± 0.57 | 10.34 ± 0.57 | 10.67 ± 0.57 | 7.34 ± 0.57 | 9.67 ± 0.57 | 7.67 ± 1.54 | ND | ND | 8.34 ± 0.57 | 9 ± 0.57 |
| *Milicia excelsa* | | | | | | | | | | |
| AE | ND | ND | ND | ND | ND | ND | ND | ND | 9.34 ± 0.57 | 8.67 ± 1.54 |
| EE | 8.67 ± 0.57 | ND | 8.34 ± 0.57 | 8.67 ± 0.57 | 9.34 ± 0.57 | 13.67 ± 0.57 | 10.34 ± 0.57 | 13.34 ± 0.57 | 8.67 ± 0.57 | 9.67 ± 0.57 |
| **Standards** | | | | | | | | | | |
| AMC | 9 ± 1 | 9.34 ± 1.15 | 17.67 ± 0.57 | 8 ± 0 | 14.67 ± 0.57 | 8 ± 0 | 8 ± 0 | 9 ± 1 | 9.34 ± 1.15 | 8 ± 0 |
| GEN | 8 ± 0 | 9.67 ± 1.57 | 18 ± 2 | 8 ± 0 | 18 ± 3.60 | 10 ± 1 | 10.34 ± 0.57 | 9.34 ± 1.52 | 9 ± 1 | 8 ± 0 |
| **Control** | | | | | | | | | | |
| Ethanol (98%) | ND | ND | ND | ND | ND | ND | ND | ND | ND | ND |
| Water | ND | ND | ND | ND | ND | ND | ND | ND | ND | ND |

Values are expressed as means ± SD; ND: not determined, EE: Ethanol extract; AE: Aqueous extract.

## 4. Discussion

The use of MPs for their pharmacological properties and biological activities is a practice that is increasingly reported across the world [65]. The WHO estimates that more than 25% of prescription drugs derive from MPs [66]. Research on and development of evidence-based phytomedicines is a priority for Africa, including Gabon [67]. Zoopharmacognosy is an original approach that could achieve these objectives [68]. Since the introduction of zoopharmacognosy as a scientific discipline, many drugs that are now in use have been found by studying animal self-medication behaviors [2]. Great apes use plants to heal themselves and thus control their parasitemia, viremia and bacteremia [69]. Moreover, traditional healers use plants items usually consumed by great apes for their pharmacological properties and biological activities [70]. However, these traditional uses of plants are not based on Western science and analysis [71].

The results in this study showed a negative correlation between extraction yield and solvent. The percentage yield of the different solvents obtained for the four plant bark extracts studied can be compared to those reported in the literature. For *Ceiba pentandra*, studies report a yield of 0.34% of the ethanolic extract [72], and 9% for the aqueous bark extract [73]. For *Myrianthus arboreus*, the yield of the ethanolic extract was 9.87% [74], and 0.05% for leaf aqueous extract [75]. For *Milicia excelsa* the yield .65% for stem bark ethanolic extract [76]. There are no reports on the yield of *Ficus* ssp. The differences observed between the results of this study and those in the literature could be explained by the many factors taken into account during the extraction process [77].

Preliminary phytochemical screening revealed the presence of several classes of secondary metabolites in the bark extracts of the four plants consumed by gorillas living in MDNP. These analyses suggest that the four plants bark investigated are good sources of natural products,

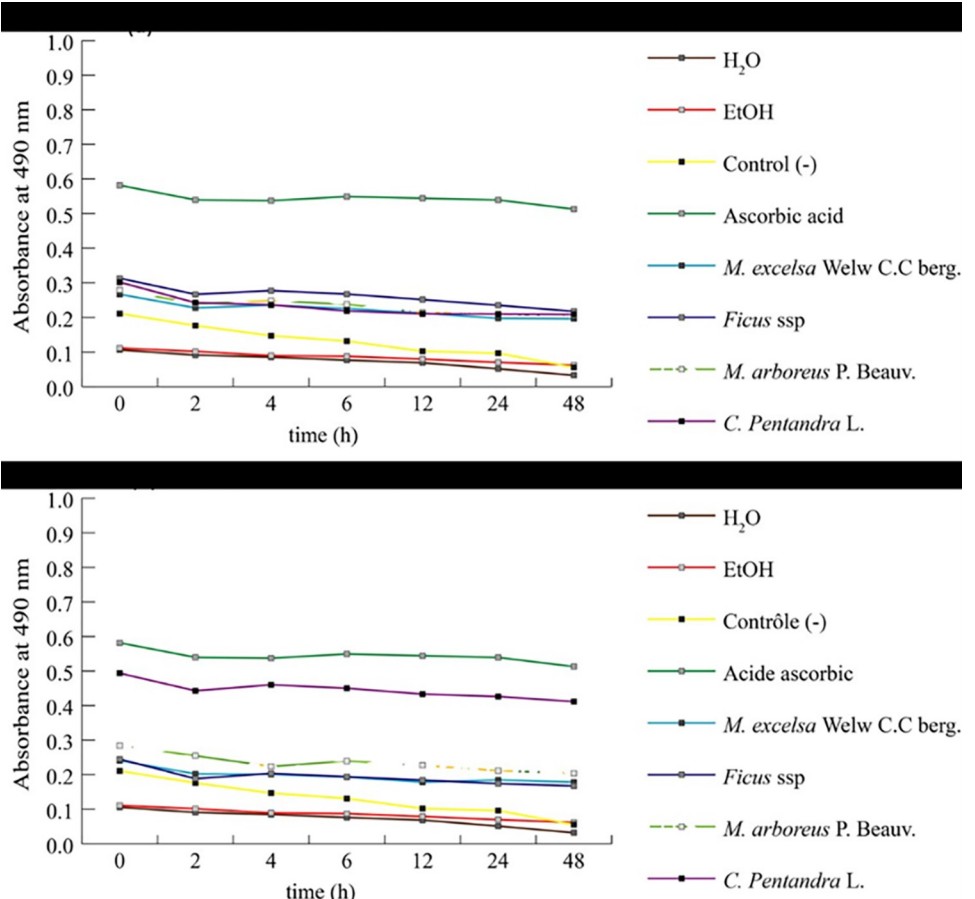

**Fig 3. β-carotene bleaching kinetics of (0.5 mg) at 490 nm from *Ceiba pentandra*, *Myrianthus arboreus*, *Ficus* ssp and *Milicia excelsa* BCE (500 μL) and Ascorbic acid.** (a): Plant aqueous BCE. (b): Plant ethanolic BCE. Each value is the mean of three analyses.

secondary metabolites endowed pharmacological properties and biological activities. Similar results have been obtained in other studies of the phytochemical content and antimicrobial activities of *C. pentandra* extracts [78, 79]. For *Myrianthus arboreus*, studies in Cameroon [80] and Ghana [81] showed comparable results to those obtained here. One study showed only the

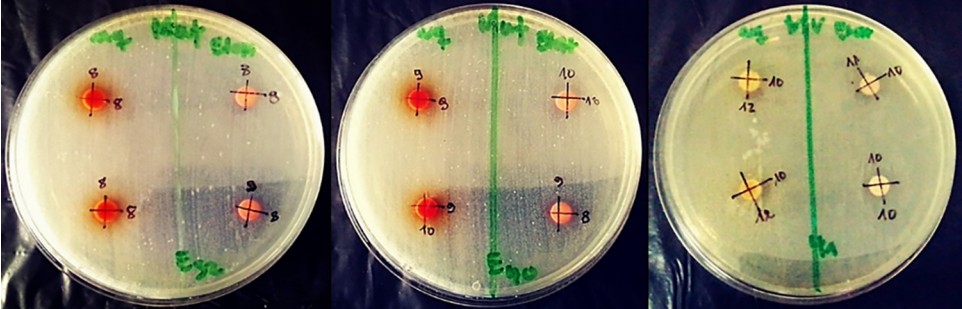

**Fig 4. Diameters of the inhibition zones (mm) produced by *Ceiba pentandra*, *Myrianthus arboreus*, *Ficus* ssp and *Milicia excelsa* BCE test against MDR *E. coli* (DECs) isolates.**

**Table 5. Minimal inhibitory (MIC) and minimal bactericidal (MBC) concentration (mg/mL) of *Ceiba pentandra*, *Myrianthus arboreus*, *Ficus* ssp and *Milicia excelsa* BCE.**

| | E32 | | E40 | | E41 | | E8 | | E4 | | E7 | | E37 | | E50 | | E49 | | E10 | |
|---|---|---|---|---|---|---|---|---|---|---|---|---|---|---|---|---|---|---|---|---|
| Sample | MIC | MBC | MIC | MBC | MIC | MBC | MIC | MBC | MIC | MBC | MIC | MBC | MIC | MBC | MIC | MBC | MIC | MBC | MIC | MBC |
| *Ceiba Pentandra* | | | | | | | | | | | | | | | | | | | | |
| AE | 12.5 | 25 | 6.25 | 12.5 | 12.5 | 25 | 12.5 | 25 | 6.25 | 12.5 | 1.56 | 3.12 | 3.12 | 6.25 | 1.56 | 3.12 | 12.5 | 25 | 6.25 | 12.5 |
| EE | 6.25 | 12.5 | 6.25 | 12.5 | 6.25 | 12.5 | 1.56 | 3.12 | 3.12 | 6.25 | 1.56 | 3.12 | 3.12 | 6.25 | 6.25 | 12.5 | 12.5 | 25 | 3.12 | 6.25 |
| *Myrianthus arboreus* | | | | | | | | | | | | | | | | | | | | |
| AE | 25 | 50 | 12.5 | 25 | — | — | — | — | — | — | — | — | — | — | — | — | — | — | — | — |
| EE | 12.5 | 25 | 6.25 | 12.5 | — | — | — | — | — | — | 3.12 | 6.25 | 3.12 | 6.25 | 3.12 | 6.25 | 3.12 | 6.25 | — | — |
| *Ficus* ssp | | | | | | | | | | | | | | | | | | | | |
| AE | — | — | — | — | 12.5 | 25 | 12.5 | 25 | 12.5 | 25 | 12.5 | 25 | 6.25 | 12.5 | 12.5 | 25 | 3.12 | 6.25 | — | — |
| EE | 6.25 | 12.5 | 3.12 | 6.25 | 3.12 | 6.25 | 25 | 50 | 6.25 | 12.5 | 25 | 50 | — | — | — | — | 12.5 | 25 | 6.25 | 12.5 |
| *Milicia excelsa* | | | | | | | | | | | | | | | | | | | | |
| AE | — | — | — | — | — | — | — | — | — | — | — | — | — | — | — | — | 6.25 | 12.5 | 12.5 | 25 |
| EE | 12.5 | 25 | — | — | 12.5 | 25 | 12.5 | 25 | 6.25 | 12.5 | 1.56 | 3.12 | 3.12 | 6.25 | 1.56 | 3.12 | 12.5 | 25 | 6.25 | 12.5 |
| Ethanol (98%) | — | — | — | — | — | — | — | — | — | — | — | — | — | — | — | — | — | — | — | — |
| Water | — | — | — | — | — | — | — | — | — | — | — | — | — | — | — | — | — | — | — | — |

MIC: minimum inhibitory concentration; MBC: minimum bactericidal concentration, EE: Ethanol extract; AE: Aqueous extract; —: Not effective.

presence of phenolic compounds and reducing sugars [82]. Another detected phenolic compounds and flavonoids [83]. *Ficus* spp are used as medicine to reduce the risk of cancer and heart disease [84]. Comparable results to those obtained here have been reported for *Ficus* species in Tunisia [85], Turkey [86] and Italy [87]. For *Milicia excelsa* BCE, a study in Nigeria, showed the presence of tannins, alkaloids, flavonoids and saponins in Nigeria [88]. Flavonoids and phenolic compounds were also found in Nigeria [89]. The secondary metabolites found in the BCE of the four plants studied possess pharmacological properties and are endowed with biological activities, suggesting that they can be used in traditional medicine as MP by healers and in pharmaceutical products [90, 91].

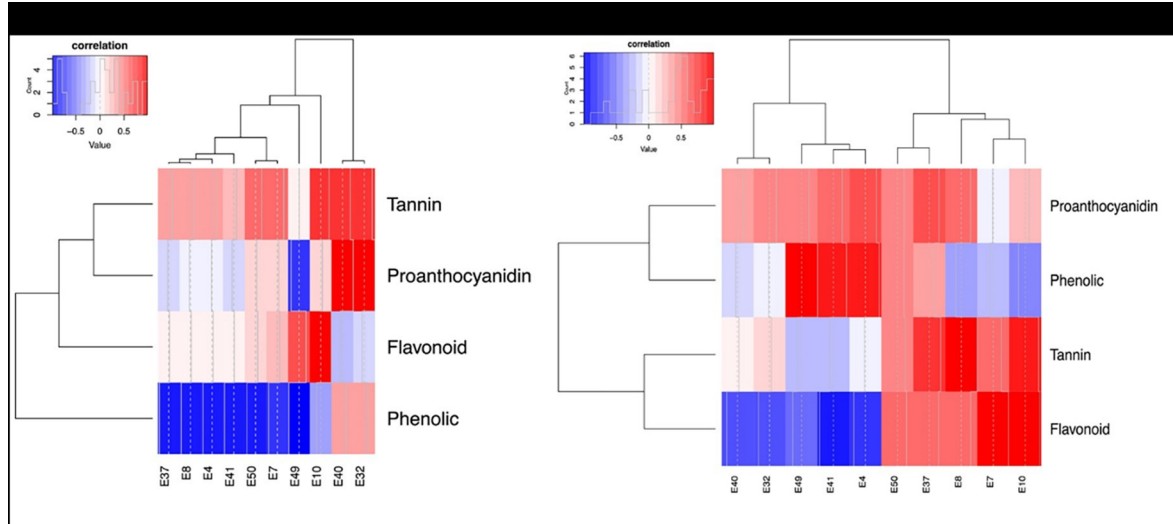

**Fig 5. Heat map illustrating the correlation between total phenolic, flavonoid, proanthocyanidin and tannin content with the assessed antimicrobial activities.** (a): Plant aqueous BCE. (b): Plant ethanolic BCE.

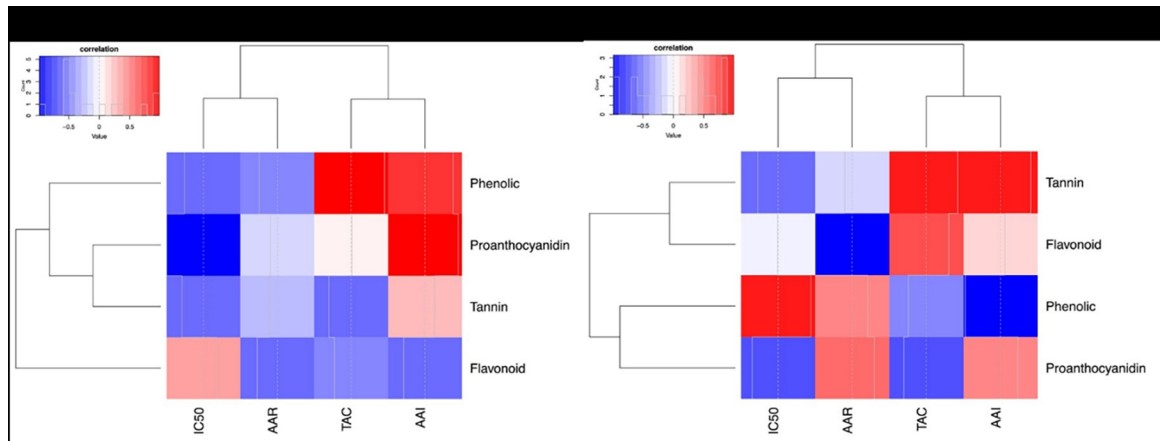

**Fig 6. Heat map illustrating the correlation between total phenolic, flavonoid, proanthocyanidin and tannin content with the assessed antioxidant activities.** (c): Plant aqueous BCE. (d): Plant ethanolic BCE.

Traditional African healers make use of MPs to treat microbial diseases without western scientific basis [92, 93]. The results for total phenolic, flavonoid and tannin content obtained here in *Ceiba pentandra* are consistent with those reported in recent studies [38, 73]. For *Myrianthus arboreus*, total phenolic and flavonoid content [94], proanthocyanidins content [95] and tannins content [81] have been found as in this study. Recent work on the genus Ficus has also revealed the presence of total phenolic compounds and total flavonoids [96], and total tannins [97]. Similar results to those obtained here have been reported for *Milicia excelsa* extracts [98]. Therefore, the use of these four plant species in traditional medicine could be attributed to the high content of phenolic [99], flavonoids (flavones and flavonones) [100], tannins [101] and proanthocyanidins [102] compounds which are known for their antimicrobial, antioxidant, anticancer, antidiarrheal, anti-inflammatory, antimalarial, anti-cytotoxic and antispasmodic activities. The phenolic, flavonoid tannin and proanthocyanidin compounds found in this study may justify their bioactive effects for diarrheal diseases, those related to oxidative stress caused by ROS and diseases related to MDR *E. coli* (DECs).

Several studies have been conducted on MPs and their extracts with the aim of determining their antioxidant capacity, total phenolic content and characterizing the respective phenolic components [103]. Most mentioned plant secondary metabolites associated with antioxidant activity are alkaloids, sulfur compounds, terpene/terpenoids, essential oil, carotene/carotenoid, polyphenol/phenol, flavonoid, tannin, and coumarin [104]. However, variations exist, even among plant species belonging to the same genus. This could be attributed to differences in the extraction method during sample preparation [105], differences in harvest time [106], differences in the variety of the analyzed sample [107], as well as differences in the climatic and soil conditions and their origins [108]. Antioxidant compounds from MPs or nutritional plants display numerous beneficial effects such as a remarkable scavenger ability against various radical species [109]. As the latter explain their action through different mechanisms, usually, various tests should be explored to fully estimate the antioxidant capacity of bark crude extracts for example [110]. Extraction of plant by solvent is a commonly used method to obtain antioxidants. However, no single solvent can extract all the secondary metabolites with antioxidant activities from plant because of its variation in solubility and polarity [111, 112]. In the present study, solvents with different polarities including water and ethanol and were used as solvents to extract antioxidants from *Ceiba pentandra* L., *Myrianthus arboreus* P. Beauv., *Ficus ssp* and *Milicia excelsa* Welw C.C berg. bark. The most natural antioxidants are

multifunctional. Therefore, a reliable antioxidant evaluation protocol requires different antioxidant activity assessments to take into account various mechanisms of antioxidant action [113].

In this study, DPPH assay showed that the radical scavenging activity of the BCEs is dose-dependent. All plant BCE studied here have DPPH radical scavenging activities. *Ficus ssp* ethanolic BCE showed a DPPH scavenging activity (IC$_{50}$) (1.31 ± 0.20 mg/mL) higher than *Ficus brevibracteata* leaves with a IC$_{50}$ value of 1.30 μg/ml (85.18 ± 0.22% of DPPH scavenging) [114], and lower than *Ficus variegata* stem bark ethanolic extract with 91% of DPPH scavenging [115]. Studies show a high relative scavenging activity (RSA) (about 50% of DPPH scavenging and 73.7 ± 8.4% of DPPH scavenging) [80, 116, 117] and *Myrianthus arboreus* aqueous extract has antioxidant activities [75]. In addition, antioxidant activities results obtained in the present study for *Myrianthus arboreus* are close to those observed in a previous study, which obtained a IC$_{50}$ value of 13.3 μg/mL for the ethanolic extract [95]. For *Ceiba pentandra* ethanolic BCE, we observed similar antioxidant activities to those in a previous study, where the DPPH radical scavenging activity of ethanolic *Ceiba pentandra* leaves extract increased in dose dependent manner ranging from 10–50 μg/ml [118]. For *Ceiba pentandra* aqueous BCE, studies have obtained comparable results from aqueous extracts of stem bark [119] which induced also a concentration-dependent radical scavenging activity on DPPH, trunk [120] and leaf bark extracts [121]. *Milicia excelsa* showed radical scavenging activity on DPPH [122, 123]. Previous studies highlighted total antioxidant activities using phosphomolybdenum total antioxidant activity assay of *Ficus nota* [124], *Ficus carica* which possessed lower reducing ability with EC$_{50}$ value of 39 μg/ml [125], *Ficus sycomorus* with EC$_{50}$ value of 25 μg/ml [125] and *Ficus Benghalensis* methanolic extract with a IC$_{50}$ value for phosphor-molybdenum of 31.84 ± 0.12 μg/ml [126]. A recent study showed *Myrianthus arboreus* aqueous root bark extract (40.3 ± 3.9 mg TE/g) and ethanolic extract (161.1 ± 11.9 mg TE/g) total antioxidant activities using phosphomolybdenum total antioxidant activity assay [95]. Sinha et al. reported *Bombax ceiba* (*Ceiba pentandra*) flowers phenolic extract total antioxidant activities using phosphomolybdenum complex assay [127]. There have been no reports on the effects of *Milicia excelsa* BCE using this assay. The β-carotene bleaching activities of all plant BCE depended on concentration. The antioxidant activity was observed in order of *Ceiba pentandra.> Myrianthus arboreus> Ficus ssp> Milicia excelsa*. There are no reports in the literature on the antioxidant activities of the BCEs of the four plants we studied using β-carotene bleaching assay. Plant antioxidants are a natural reservoir of bioactive compounds and play important roles in plant acclimation and adaptation to environmental challenges, and are also beneficial to human health [104]. Recent work on pollen extracts of three plants from the Palestinian pharmacopoeia reported antioxidant activities of flavonoids compounds such as flavones and flavonols in these extracts [48]. The antioxidant capacities of these all secondary compounds support the human body's battle against diseases by absorbing free radicals and chelating metal ions that could catalyze the production of ROS, which facilitates lipid peroxidation [128]. Antioxidants break radical chain reactions, preventing oxidative stress-related damage [129]. In brief, antioxidants can act according to two major mechanisms, either by transfer of hydrogen atom or by electron transfer [130].

However, in this study, regarding the three methods used (DPPH radical capacity and Phosphomolybdenum complex for total antioxidant capacity and β-Carotene bleaching assay), ethanolic extracts of *Ceiba pentandra* L., *Ficus ssp* and *Milicia excelsa* Welw C.C berg. excepted for *Myrianthus arboreus* P. Beauv., presented relatively higher antioxidant activity than the aqueous ones. In a practical way, plant extracts antioxidant activity depends on several factors, for example: the concentration of the extracts, the method of evaluation, the sensitivity of the antioxidants to the temperature of the test, and the water- or fat-soluble nature of the antioxidant [131, 132].

Regarding antimicrobial activity, previously published reports report that the inhibitory activity of MPs extracts against Gram-positive and Gram-negative bacteria has been widely highlighted in the literature [133], and that MPs extracts with MIC values less than 100 mg/mL can be considered to have very good antimicrobial activity [134, 135]. In addition, the antimicrobial activity of these MPs extracts could be different toward various kinds of bacteria and different kinds of extract [136, 137]. This can be explained by the highest resistance of Gram-negative bacteria due to the complexity of their cell wall, containing a double membrane as opposed to the unique glycoprotein/teichoic acid membrane of Gram-positive bacteria [138]. In light of these indications, the antimicrobial activity of the bark crude extracts from the four plants studied and consumed by western lowland gorillas, against MDR *E. coli* (DECs) isolates, varied with different extraction solvents.

Results of the antimicrobial activities from the four plant BCEs investigated against MDR *E coli* (DECs), found in this study are comparable to those reported in the literature. *Ceiba pentandra* ethanolic extract showed antimicrobial activities against *E. coli* (an organism frequently implicated in gastroenteritis and pelvic inflammation) with a zone of inhibition value of $10.55 \pm 1.45$ mm and showed a MIC value of 12.5 mg/ml against *E. coli* [72]. A previous study also showed that *Ceiba pentandra* ethanolic extract was highly active against *E. coli* and that the antimicrobial activity of leaf extract increased as the concentration of ethanolic extract of *Ceiba pentandra* leaves increased [118]. Parulekar et al. [78] reported moderate antimicrobial activities from aqueous extract and strong antimicrobial activities ethanolic extract against *E. coli*, which increased as the concentration of the extracts increased; and Njokuocha et al. [79] also reported antimicrobial activities of *Ceiba pentandra* extracts. *Ceiba pentandra* aqueous extract has shown antimicrobial activities against Gram-negative bacteria, including *E. coli* [78, 79]. A recent study showed that the mean zone of inhibition of *Myrianthus arboreus* was zero for *E. coli* ATCC 25922 with aqueous and ethanolic extracts. However, for clinical MDR isolates, the *Myrianthus arboreus* inhibition zone diameter value was 0.4–2.2 mm for ethanolic extracts, showing low antimicrobial activities [82]. Ethanolic extracts of taxa belonging to the genus Ficus have been studied for their antimicrobial potential [139]. Biologically, *Milicia excelsa* Welw C.C berg. extract antimicrobial activities against enterobacteria have been demonstrated [44]. The flavonoid neocyclomorusin isolated from *Milicia excelsa* extract exhibited antimicrobial activity against *K. pneumoniae* ATCC11296 and *E. cloacae* BM47, with MIC values of 4 µg/mL each [43]. Padayachee et al. reported zero antimicrobial activity of *Milicia excelsa* extract against *E. coli* [140].

Many of these bioactive compounds are believed to have been used by plants and their parts, during their evolution, to protect against bacteria and are responsible for antimicrobial activity [141]. A possible mechanism for this phytochemical activity may be either through inhibiting the growth of microbes, inducing cellular membrane perturbations, interference with certain microbial metabolic processes, or modulation of signal transduction or gene expression pathways. However, these mechanisms may all occur at the same time as a result of the synergistic effect between the compounds [142]. The roles of secondary metabolites are relatively straightforward; for instance, they participate in general protective roles (antioxidant, free radical scavenging, UV light absorbing, and antiproliferative agents) and protect the plant from herbivorous animals (grazing) including different pathogenic microorganisms such as bacteria, fungi, and viruses. They also manage interplant relationships, acting as allelopathic defenders of the plant's growing space against competitor plants [143].

Several phytochemicals have already been identified using GC-MS or HPLC-MS for *Ceiba pentandra* [118, 120], for *Myrianthus arboreus* [41, 95] and for *Milicia excelsa* [43]. These bioactive components are known to exhibit medicinal property [144]. In addition, in the current study, a positive correlation was observed between total phenolic, flavonoid, proanthocyanidin

and tannin content with antimicrobial and antioxidant activity. The antioxidant activity of an extract or compound is often associated with their redox proprieties, which allow them to act as reducing agents [145]. Several other studies have also reported positive correlations between plant extracts secondary metabolites such as phenolic [146], flavonoid [147], proanthocyanidin [148], tannin [149] and antimicrobial activity.

Biodiversity contributes significantly towards human livelihood and development and thus plays a predominant role in the well-being of the global population [150]. Valorization of plants with medicinal value is a world challenge that meets the objectives of biodiversity conservation [104, 151]. The latter could involve the study of phytochemistry, pharmacological properties, antioxidant and antimicrobial activities of plants consumed by non-human animals, including great apes such as western lowland gorilla [2, 152]. Additionally, new drug discovery from natural sources involve a multifaceted approach combining botanical, phytochemical, biological, and molecular techniques [153, 154]. These bioprospecting practices have implications for medicine, environment, economy, public health, and culture [155]. Unfortunately, the potential benefits of plant-based medicines have led to unscientific exploitation of natural resources, a phenomenon that is being observed globally. The decline in biodiversity and loss due to species extinction is largely the result of the rise in the global population, rapid and sometimes unplanned industrialization, indiscriminate deforestation, overexploitation of natural resources, illegal trade, pollution, and finally global climate change [155, 156]. Therefore, it is of utmost importance that biodiversity is preserved, to provide future structural diversity and compounds for the sustainable development of human civilization [153, 157]. This becomes even more important for low and middle-income nations, where well-planned bioprospecting coupled with non-destructive commercialization could help in the conservation of biodiversity, ultimately benefiting humankind in the long run [153, 157].

## 5. Conclusion

The results of this study, which examined the antioxidant and antimicrobial activity of *Ceiba pentandra*, *Myrianthus arboreus*, *Ficus* ssp and *Milicia excelsa* BCEs, plants are consumed by western lowland gorilla living in MDNP and used in traditional medicine by Gabonese healers, revealed some important facts. Indeed, all plant BCEs studied showed antioxidant and antimicrobial activities. The asymptomatic nature of theses gorillas with regard MDR *E. coli* (DECs) could be explained by their consumption of the bark of the four plants tested. The scientific results obtained during pharmacological analyses could justify the use of these plants in the traditional pharmacopoeia against various human diseases. The BCEs of the four plants studied could be promising sources for new bioactive molecules discovery in the pharmaceutical, cosmetics and food industries. One of the potential challenges of this study was to address the issue of potential alternative solutions to the problem of antimicrobial resistance, using a zoo-pharmacognosy approach.

These results show that the BCE of these plants could be used as an effective treatment for diseases caused by free radicals and diseases caused by antimicrobial-resistant bacterial strains. Then, all this founding could comfort the self-medication hypothesis of non-human animals, including great apes. The results of our study suggest that all plant BCEs studied could potentially be candidate improved traditional medicines (ITMs) in the application of new therapeutic protocols against infectious diseases of bacterial origin.

In addition, the identification of all plant BCEs studied bioactive compounds, using HPLC-MS or a LC-MS/MS and molecular network approach, would add quality to the results obtained in our study. This approach is a valuable tool for revealing the metabolomes of plants extracts, groups secondary metabolites into molecular families based on their spectral

similarities, and identify known compounds in order to focus on unknown compounds that may potentially be of biological interest.

## Supporting information

**S1 Checklist. Inclusivity in global research.**
(DOCX)

**S1 File.**
(DOCX)

## Acknowledgments

The authors would like to acknowledge Doctoral School of Tropical Infectious Diseases of Franceville (EDR), Interdisciplinary Center for Medical Research of Franceville (CIRMF) and University of Sciences and Technology of Masuku (USTM) of Franceville. We also acknowledge Jean Bernard LEKANA-DOUKI and Jacques LEBIBI, Managing, Christophe NGOKO-MAKA, mission vehicle driver and Peter MOMBO, Doussala station trackers team leader and all his collaborators.

## Author Contributions

**Conceptualization:** Leresche Even Doneilly Oyaba Yinda, Richard Onanga, Etienne François Akomo-Okoue, Sylvain Godreuil.

**Data curation:** Leresche Even Doneilly Oyaba Yinda.

**Formal analysis:** Leresche Even Doneilly Oyaba Yinda, Cédric Sima Obiang, Judicaël Obame-Nkoghe.

**Investigation:** Leresche Even Doneilly Oyaba Yinda, Roland Mitola.

**Methodology:** Leresche Even Doneilly Oyaba Yinda, Richard Onanga, Cédric Sima Obiang, Herman Begouabe, Roland Mitola.

**Project administration:** Leresche Even Doneilly Oyaba Yinda, Cédric Sima Obiang.

**Resources:** Etienne François Akomo-Okoue, Judicaël Obame-Nkoghe, Joseph-Privat Ondo, Guy-Roger Ndong Atome, Louis-Clément Obame Engonga, Sylvain Godreuil.

**Software:** Leresche Even Doneilly Oyaba Yinda.

**Supervision:** Ibrahim.

**Validation:** Leresche Even Doneilly Oyaba Yinda, Richard Onanga, Sylvain Godreuil.

**Visualization:** Leresche Even Doneilly Oyaba Yinda, Richard Onanga, Sylvain Godreuil.

**Writing – original draft:** Leresche Even Doneilly Oyaba Yinda.

**Writing – review & editing:** Leresche Even Doneilly Oyaba Yinda, Richard Onanga, Cédric Sima Obiang, Joanna M. Setchell.

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
