## [Decision Letter · Decision Letter 0]

11 Jan 2024

PONE-D-23-26722Antibacterial Activities of Four Antioxidant Plants Consumed by Western Lowland Gorilla (Gorilla gorilla gorilla) living in Moukalaba-Doudou National Park (MDNP) infected by MDR E. coli (DECs).PLOS ONE

Dear Dr. OYABA YINDA,

Thank you for submitting your manuscript to PLOS ONE. After careful consideration, we feel that it has merit but does not fully meet PLOS ONE’s publication criteria as it currently stands. Therefore, we invite you to submit a revised version of the manuscript that addresses the points raised during the review process.

Please see the comments from five reviewers below and in the attachments. Several of the reviewers have requested that the framing of the study is enhanced and that the contribution in light of previous work is highlighted. One reviewer touched on the fact that an LC-MS analysis of the main secondary metabolites may have strengthened the study - however, we recognise that this may be outside the scope of the current study. If you have these data, however, please consider incorporating these.

We look forward to receiving your revised manuscript.

Kind regards,

Hanna Landenmark

Staff Editor

PLOS ONE

Journal Requirements:

3. Please include a complete copy of PLOS’ questionnaire on inclusivity in global research in your revised manuscript. Our policy for research in this area aims to improve transparency in the reporting of research performed outside of researchers’ own country or community. The policy applies to researchers who have travelled to a different country to conduct research, research with Indigenous populations or their lands, and research on cultural artefacts. The questionnaire can also be requested at the journal’s discretion for any other submissions, even if these conditions are not met. Please find more information on the policy and a link to download a blank copy of the questionnaire here: https://journals.plos.org/plosone/s/best-practices-in-research-reporting. Please upload a completed version of your questionnaire as Supporting Information when you resubmit your manuscript.

"Non.

Les bailleurs de fonds n'ont joué aucun rôle dans la conception de l'étude, la collecte et l'analyse des données, la décision de publier ou la préparation du manuscrit."

5. We note that your Data Availability Statement is currently as follows: [Toutes les données pertinentes se trouvent dans le manuscrit et ses fichiers d'informations complémentaires.]

6. In the online submission form, you indicated that your data will be submitted to a repository upon acceptance.  We strongly recommend all authors deposit their data before acceptance, as the process can be lengthy and hold up publication timelines. Please note that, though access restrictions are acceptable now, your entire minimal  dataset will need to be made freely accessible if your manuscript is accepted for publication. This policy applies to all data except where public deposition would breach compliance with the protocol approved by your research ethics board. If you are unable to adhere to our open data policy, please kindly revise your statement to explain your reasoning and we will seek the editor's input on an exemption. 

7. Please note that in order to use the direct billing option the corresponding author must be affiliated with the chosen institute. Please either amend your manuscript to change the affiliation or corresponding author, or email us at plosone@plos.org with a request to remove this option.

8. We note that Figure 1 in your submission contain map/satellite image which may be copyrighted. All PLOS content is published under the Creative Commons Attribution License (CC BY 4.0), which means that the manuscript, images, and Supporting Information files will be freely available online, and any third party is permitted to access, download, copy, distribute, and use these materials in any way, even commercially, with proper attribution. For these reasons, we cannot publish previously copyrighted maps or satellite images created using proprietary data, such as Google software (Google Maps, Street View, and Earth). For more information, see our copyright guidelines: http://journals.plos.org/plosone/s/licenses-and-copyright.

Reviewers' comments:

Reviewer's Responses to Questions

**Comments to the Author**

1. Is the manuscript technically sound, and do the data support the conclusions?

Reviewer #1: Yes

Reviewer #2: Yes

Reviewer #3: Partly

Reviewer #4: Yes

Reviewer #5: Partly

2. Has the statistical analysis been performed appropriately and rigorously? 

Reviewer #1: Yes

Reviewer #2: Yes

Reviewer #3: Yes

Reviewer #4: I Don't Know

Reviewer #5: Yes

3. Have the authors made all data underlying the findings in their manuscript fully available?

Reviewer #1: Yes

Reviewer #2: Yes

Reviewer #3: Yes

Reviewer #4: Yes

Reviewer #5: No

4. Is the manuscript presented in an intelligible fashion and written in standard English?

Reviewer #1: Yes

Reviewer #2: Yes

Reviewer #3: No

Reviewer #4: Yes

Reviewer #5: Yes

5. Review Comments to the Author

Reviewer #1: There are some Comments for the Conclusion:

The conclusion provides valuable information about the antioxidant and antimicrobial properties of the studied plant extracts and their potential applications. However, the text could benefit from improved organization and clarity. Consider breaking down the conclusion into subsections to address different aspects more coherently, such as the results, applications, future research, and potential challenges.

The conclusion mentions the need for in vivo studies to understand the safety and efficacy of the plant extracts. It would be valuable to specify what types of safety and efficacy assessments are required and to discuss any potential challenges or ethical considerations in conducting such studies.

The conclusion suggests the need for the isolation and identification of bioactive compounds from the plant extracts. Expanding on this point by discussing potential methods for isolation and identification, as well as the significance of these compounds, would provide a more comprehensive outlook.

The text mentions using these plant extracts as substitutes for commercially available synthetic drugs. It would be beneficial to elaborate on which specific diseases or conditions these extracts might address and the potential advantages over synthetic drugs, such as reduced side effects.

While the conclusion discusses the pharmaceutical, cosmetic, and food industries as potential applications, it could delve into the specific products or treatments that might benefit from these plant extracts and how they could be incorporated.

Reviewer #2: The manuscript “Phytochemical Screening, Antioxidant and Antibacterial 1 Activities of Four Plants Consumed by Western Lowland Gorilla (Gorilla gorilla gorilla) living in Moukalaba-Doudou National Park (MDNP).” written by YINDA et al is well prepared and could be a significance contribution in the plant taxonomy. The manuscript can be considered for publication after addressing these minor corrections.

All corrections are in the pdf file as track changes.

Reviewer #3: -The paper is too long. It has to be shortened and stick only on essential points.

-2.3.1 extraction lines 148-149

This is not a new extraction procedure so figure 2 is without any added value. In addition Figure 2 is cited within the text as figure 1, see line 147.

-Line 271 Table 1. Caption: plant species……including diarrhoea…. What is the need to highlight diarrhoea here? And also diarrhoea is not listed as an indication for none of the plant species in the table.

-Order the plants in the table in alphabetical order either using the family name or the species name for the table to be more searchable. It could also be interesting to give more details about the mode of preparation.

-Line 275-276. Remove Table 2. Most of the data in table 2 are already listed in Table 1. Apart from the medicinal applications which actually refer to previous biological activities of plants extracts demonstrated in the literature.

-Remove Table 3. I believe the extraction yield id not a significant result needing a whole table of its own. This can be move to the supplementary information. A short sentence can be used to describe this result within the document.

-Table 4 biological activity coulumn: What does it stand for?

-Lines 298-310 are repeating the data described in the figure 3a and 3b. I suggest to delete this part. And add calibration curves to the supplementary information.

-What does the black rectangle in the legend of figures 3a and 3b stand for?

-Line 336 figure 4 instead of figure 3

Reviewer #4: Title: The title must be reformulate ".....activities of Aqueous and Ethanolic Extracts of Four plants......."

Abstract:

The First sentence of the abstract, is the only definition of Zoopharmacology and not resume the background and Justification of the Study

Introduction line 72:This sentense must be reformulate, ROS not produced Cancer, they can be responsible or contribute to Cancer development

Concerning the review, the authors are interesting to study the antibacterial and antioxidant properties of extracts from 4 plants consumed by gorilla for their health being. With the aim of discovery new drugs. But the authors should revised the justification of the choice of these four plants. As well as, there is a lot of scientific work that have been already done on these plants, the review should be update.

Line 144: Extraction

The nature of treatment of plant material is

The sentence should be reformulate, because after filtration of the two (ethanolic and aqueous) maceration, filtration is performed on the two and aqueous extract is direct freeze-dried, it is only ethanolic extract which is concentrate and should be dry in an oven

Moreover, why the choice of Water and ethanol only as solvent?

Line 179: DPPH radical scavenging activity

Why each of the eight BCE (particularly aqueous) were add to methanol to obtains stock solution? Why not in its solvent extraction solvent? What were containing in Blank Tube? Give details

Line 174: Table 1

The biological activities of the four plants always demonstrated should be resume in the review, also their utilization to justify their choice.

Figure 3a, 3b: What represent the black Rectangle, It should be remove on the figure

Conclusion:

According all the experimentation performed during the study. It not possible to choice the plant which presents the best activities to pursue the study (in vivo experimentation, isolation, identification and Characterization of bioactive compounds presents in plants)?

Reviewer #5: The authors should include a problem statement, motivation as well as hypothesis in the introduction. It will be difficult to discussion without a clear problem being addressed. The conclusion will be easier to reach if there is appropriate alignment with the problem statement

6. PLOS authors have the option to publish the peer review history of their article (what does this mean?). If published, this will include your full peer review and any attached files.

Reviewer #1: No

Reviewer #2: No

Reviewer #3: No

Reviewer #4: No

Reviewer #5: **Yes: **Buhlebelive Mndzebele

---

## [Author Response · Author response to Decision Letter 0]

3 Feb 2024

Reviewer #1: There are some Comments for the Conclusion:

The conclusion provides valuable information about the antioxidant and antimicrobial properties of the studied plant extracts and their potential applications. However, the text could benefit from improved organization and clarity. Consider breaking down the conclusion into subsections to address different aspects more coherently, such as the results, applications, future research, and potential challenges.

Done

The conclusion mentions the need for in vivo studies to understand the safety and efficacy of the plant extracts. It would be valuable to specify what types of safety and efficacy assessments are required and to discuss any potential challenges or ethical considerations in conducting such studies.

This part of the conclusion has been reworded. Indeed, when we spoke of in vivo testing, we were referring to the study of the cytotoxicity of extracts, by assessing their effects on the overall cell proliferation of cell lines.

The conclusion suggests the need for the isolation and identification of bioactive compounds from the plant extracts. Expanding on this point by discussing potential methods for isolation and identification, as well as the significance of these compounds, would provide a more comprehensive outlook.

This part of the conclusion has been reworded as recommended by the reviewer.

Molecular networking (MN) is a valuable tool for revealing the metabolomes of plants, humans, microorganisms and animals [1]. It can not only annotate compounds in complex matrices based on their tandem mass spectrometry (MS/MS) characteristics but also group them into molecular families based on their spectral similarities, thus facilitating visualization of structurally identical molecules within the same chemical family and structurally different molecules distributed in different chemical families [2, 3]. This approach allows researchers to identify known compounds in order to focus on unknown compounds that may potentially be of biological interest [4].

The text mentions using these plant extracts as substitutes for commercially available synthetic drugs. It would be beneficial to elaborate on which specific diseases or conditions these extracts might address and the potential advantages over synthetic drugs, such as reduced side effects.

Done

While the conclusion discusses the pharmaceutical, cosmetic, and food industries as potential applications, it could delve into the specific products or treatments that might benefit from these plant extracts and how they could be incorporated.

We plan to develop this aspect of our research, focusing on the potential represented by improved traditional medicines (ITMs).

Reviewer #2: The manuscript “Phytochemical Screening, Antioxidant and Antibacterial 1 Activities of Four Plants Consumed by Western Lowland Gorilla (Gorilla gorilla gorilla) living in Moukalaba-Doudou National Park (MDNP).” written by YINDA et al is well prepared and could be a significance contribution in the plant taxonomy. The manuscript can be considered for publication after addressing these minor corrections.

The authors have done great research. However, more information and or explanation is required

1. Please include the references in line 62-63

Done

2. Are the items listed in 67-70 going to be part of the study. If not, please revise.

Done

3. The introduction lacks the problem statement and motivation to link the chosen species and the natural medicine. Provide more information and compelling reasons for the audience to clearly understand.

Done

4. Please also include the hypothesis

Done

5. In the methodology section 

a. Please state if your collection methodology has similarities with the manner in which gorillas consume. It would be interesting to understanding the rationale behind the sampling methodology or approach

Done

Sampling of plant barks was guided by the fact that these were not only consumed by gorillas but also used in traditional medicine by traditional practitioners healers, based on a non-invasive observation and collection methodology.

b. Fig 3 a, b; would be better visible with whole numbers

Done

c. Fig 4; why a different font?

Changes have been made to correct the fonts in this figure.

6. Line 265; there is mention of other plant organs of interest which have not been stated in the introduction or methodology. Any reasons to mention these at this stage?

There was no particular reason to mention other interesting plant organs not mentioned in the introduction or methodology. This was done for information purposes, in order to highlight the various other organs and preparation processes (maceration, decoction, infusion, etc.) used by local populations in their traditional medicine, which gorillas cannot do.

7. The discussion does not explain results except to relate it to other studies. Explain why certain observations as well as the reasons behind the observsations

Done

Reviewer #3: -The paper is too long. It has to be shortened and stick only on essential points.

-2.3.1 extraction lines 148-149

This is not a new extraction procedure so figure 2 is without any added value. In addition Figure 2 is cited within the text as figure 1, see line 147.

This figure has been removed from the document.

-Line 271 Table 1. Caption: plant species……including diarrhoea…. What is the need to highlight diarrhoea here? And also diarrhoea is not listed as an indication for none of the plant species in the table.

Diarrhoea was highlighted here as it could potentially define the pathological state of the gorillas in this study, given the results obtained in a previous study [5], revealing that these animals were harbouring MDR E. coli (DECs), potentially pathogenic to humans. Moreover, bacterial diarrhoea is still a major public health threat, for which the search for new antimicrobial drugs remains necessary [6].

-Order the plants in the table in alphabetical order either using the family name or the species name for the table to be more searchable. It could also be interesting to give more details about the mode of preparation.

Done

-Line 275-276. Remove Table 2. Most of the data in table 2 are already listed in Table 1. Apart from the medicinal applications which actually refer to previous biological activities of plants extracts demonstrated in the literature.

Table 2 has been removed from the document, but like other elements of the manuscript will be inserted in an additional file.

-Remove Table 3. I believe the extraction yield id not a significant result needing a whole table of its own. This can be move to the supplementary information. A short sentence can be used to describe this result within the document.

Done

-Table 4 biological activity coulumn: What does it stand for?

This expression has been reworded. Our intention was simply to highlight the fact that the various classes of secondary metabolites found in the plant extracts studied had pharmacological activities, by way of illustration, the list of which for each of the families was non-exhaustive.

-Lines 298-310 are repeating the data described in the figure 3a and 3b. I suggest to delete this part. And add calibration curves to the supplementary information.

Done

-What does the black rectangle in the legend of figures 3a and 3b stand for?

As this rectangle had no particular significance, it has been removed from these figures.

-Line 336 figure 4 instead of figure 3

Done

Reviewer #4: Title: The title must be reformulate ".....activities of Aqueous and Ethanolic Extracts of Four plants......."

Done

Abstract:

The First sentence of the abstract, is the only definition of Zoopharmacology and not resume the background and Justification of the Study

This part of the abstract has been revised.

Introduction line 72:This sentense must be reformulate, ROS not produced Cancer, they can be responsible or contribute to Cancer development

This sentence has been reworded.

Concerning the review, the authors are interesting to study the antibacterial and antioxidant properties of extracts from 4 plants consumed by gorilla for their health being. With the aim of discovery new drugs. But the authors should revised the justification of the choice of these four plants. As well as, there is a lot of scientific work that have been already done on these plants, the review should be update.

The choice of these plants for the study was guided by a study carried out on this site concerning a certain number of plants consumed by the gorillas of the PNMD, but above all by the results of ethnobotanical and ethnopharmacological surveys carried out among the local population, on the uses of these plants in traditional medicine, for the treatment of human illnesses.

A review of recent studies carried out on these plants is presented in Table 2, which will now be added to the supplementary file on the recommendation of a reviewer.

Line 144: Extraction

The nature of treatment of plant material is

The sentence should be reformulate, because after filtration of the two (ethanolic and aqueous) maceration, filtration is performed on the two and aqueous extract is direct freeze-dried, it is only ethanolic extract which is concentrate and should be dry in an oven

This sentence has been reworded.

Moreover, why the choice of Water and ethanol only as solvent?

Of course, there are many other solvents that could have been used for extractions, but unfortunately we only had these two reagents in our laboratory.

Line 179: DPPH radical scavenging activity

Why each of the eight BCE (particularly aqueous) were add to methanol to obtains stock solution? Why not in its solvent extraction solvent? What were containing in Blank Tube? Give details

Answer: The addition of methanol to obtain a stock solution was carried out in order to optimize miscibility with DPPH, bearing in mind that 400 μl of aqueous solvent had previously been used to prepare the aqueous extract solutions for each plant.

For the aqueous extracts, the blank tube contained 400 μl of water and 1.6 ml of methanol.

Line 174: Table 1

The biological activities of the four plants always demonstrated should be resume in the review, also their utilization to justify their choice.

The biological activities of the four plants always demonstrated, as well as their use, have been listed in a table added to the supplementary file on the recommendation of one of the reviewers.

Figure 3a, 3b: What represent the black Rectangle, It should be remove on the figure

As this rectangle had no particular significance, it has been removed from these figures.

Conclusion:

According all the experimentation performed during the study. It not possible to choice the plant which presents the best activities to pursue the study (in vivo experimentation, isolation, identification and Characterization of bioactive compounds presents in plants)?

If we rely solely on the overall results obtained during our study, we cannot clearly recommend any of these four plants for the above-mentioned studies, until we have carried out more advanced studies such as identifying the bioactive compounds in the crude bark extracts of the four plants tested using the HPLC-MS method or using an LC-MS/MS and molecular network approach, in order to determine their real potential.

However, Ficus ssp would be of particular interest, given that little or no study of its biological and pharmacological activities has been carried out to date.

Reviewer #5: The authors should include a problem statement, motivation as well as hypothesis in the introduction. It will be difficult to discussion without a clear problem being addressed. The conclusion will be easier to reach if there is appropriate alignment with the problem statement

A problem statement, motivation and hypothesis have been included in the introduction.

---

## [Decision Letter · Decision Letter 1]

26 Mar 2024

PONE-D-23-26722R1Activités des extraits aqueux et éthanoliques de quatre plantes antioxydantes consommées par le gorille des plaines occidentales ( Gorilla gorilla gorilla ).PLOS ONE

Dear Dr. OYABA YINDA,

Thank you for submitting your manuscript to PLOS ONE. After careful consideration, we feel that it has merit but does not fully meet PLOS ONE’s publication criteria as it currently stands. Therefore, we invite you to submit a revised version of the manuscript that addresses the points raised during the review process. **ACADEMIC EDITOR**: Please revise your article title both in the submission system and in the article to be in English/the same.The defintion of Zoopharmacognosy in the main text may not be right. To my knowledge, this has to do with non-human animals using natural products for self medicating. Please recheck.In Table 2, what basis is used to indicate the abundance of the phytochemicals? Is it intensity of colors/foam oberved?. This should be mentioned explicitly in the legend.There are so many unnecessary in-text citations. I have offered some suggestions in the attached draft file to allow for deletion of some that may not be useful.Please summarize your CONCLUSIONS.References should have complete bibliographic information required. Please refer to the journal guidelines. Check the intext citations, some references do not match. For example, Wubetu et al. [48].  does match reference [48] which is Biwôle et al., Iroko wood (Milicia excelsa CC berg), a good candidate for high-speed rotation-induced wood dowel welding: An assessment of its welding potential and the water resistance of its welded joints. International Journal of Adhesion and Adhesives, 2023. 123: p. 103360

We look forward to receiving your revised manuscript.

Kind regards,

Timothy Omara, PhD

Academic Editor

PLOS ONE

Journal Requirements:

1. Please complete all items on the Human Participants Research Checklist that are relevant for your submission, by following this link:  http://journals.plos.org/plosone/s/file?id=dc11/PLOSOne_Human_Subjects_Research_Checklist.docx (Contact us at plosone@plos.org if you cannot access the document.) There may be overlap between the checklist items and other queries listed below; please address any duplicated queries both in your response email and on the checklist itself. Upload the completed Human Participants Research Checklist as file type “Other” when you re-submit your manuscript. This document is for internal journal use only and will not be published if your article is accepted. The requested information will help us to assess whether your submission complies with PLOS ONE’s policies and adheres to applicable reporting standards. Note that your manuscript may be rejected if you provide incomplete or inadequate responses to the checklist questions and that changing the ‘Section/Category’ of your article does not affect this requirement.

Reviewers' comments:

Reviewer's Responses to Questions

**Comments to the Author**

1. If the authors have adequately addressed your comments raised in a previous round of review and you feel that this manuscript is now acceptable for publication, you may indicate that here to bypass the “Comments to the Author” section, enter your conflict of interest statement in the “Confidential to Editor” section, and submit your "Accept" recommendation.

Reviewer #1: All comments have been addressed

Reviewer #5: All comments have been addressed

2. Is the manuscript technically sound, and do the data support the conclusions?

Reviewer #1: Yes

Reviewer #5: Partly

3. Has the statistical analysis been performed appropriately and rigorously? 

Reviewer #1: Yes

Reviewer #5: Yes

4. Have the authors made all data underlying the findings in their manuscript fully available?

Reviewer #1: (No Response)

Reviewer #5: No

5. Is the manuscript presented in an intelligible fashion and written in standard English?

Reviewer #1: (No Response)

Reviewer #5: Yes

6. Review Comments to the Author

Reviewer #1: The authors addressed all comments. There are no new comments, and it can be accepted for publication in the journal.

Reviewer #5: Dear Author

Title: Antibacterial Activities of Aqueous and Ethanolic Extracts of Four Antioxidant Plants 2 Consumed by Western Lowland Gorilla (Gorilla gorilla gorilla).

Aim: The aim of this study was to determine the antioxidant and antibacterial activity of Ceiba pentandra, Myrianthus arboreus, Ficus ssp. and Milicia excelsa bark crude extracts (BCE), plants consumed by western lowland gorillas (Gorilla gorilla gorilla) and used in traditional medicine, and then to characterize their phytochemical compounds.

I have gone through the document.

These are my suggestions

In the introduction;

Please focus on the species of concern in relation to the study.

Its usage, values, usage and link everything to Gorillas

The desired medicinal properties should be aligned to the species of concern

I would like to believe that your study focuses on specific species. These should be addressed in the introduction.

Are the aspects mentioned from line 77 to 82 part of the study?

Once again try and bring focus to your study in relation to the aspects of interest as stated in the objectives

Characterization as well as taxonomy, GPS coordinates of the species of interest is key

There is Table 1 which has the biggest list. There after we have table 2 with fewer species. There should a clear flow of information on the reasons. Otherwise you can just stick to the species of interest only and justify rather than having species that are not part of the study

Discussion

State general observations

What caused those observations (positive and negative factors)

Compare with others, preferably recent

7. PLOS authors have the option to publish the peer review history of their article (what does this mean?). If published, this will include your full peer review and any attached files.

Reviewer #1: No

Reviewer #5: **Yes: **Buhlebelive Mndzebele

---

## [Author Response · Author response to Decision Letter 1]

23 Jun 2024

ACADEMIC EDITOR:

Please revise your article title both in the submission system and in the article to be in English/the same.

Done

The defintion of Zoopharmacognosy in the main text may not be right. To my knowledge, this has to do with non-human animals using natural products for self medicating. Please recheck.

Done

MDNP

This was difinided

The animals using these species are critically endangered according to the latest assessment (https://doi.org/10.2305%2FIUCN.UK.2016-2.RLTS.T9406A136251508.en).

We didn't need a special permit for this sampling because of the elements presented in the document, but also because we used the non-invasive method as previously discribed [1] to take these plant bark samples.

ssp

In taxonomy, in the Latin naming of living beings, the name of subspecies may be accompanied by the abbreviation ssp., which is that of the Latin term subspecies [2].

This may not be useful. Simply use a standard antioxidant like ascorbic acid and compare the IC50 values

This part have been removed from the document

If you wish to include this, then describe the procedures for the ethnobotanical survey. Also, your title may need to reflect it

The ethnobotanical survey procedures are described in the following sections: 2.2. Data and sample collection and Therapeutic indications

In Table 2, what basis is used to indicate the abundance of the phytochemicals? Is it intensity of colors/foam oberved?. This should be mentioned explicitly in the legend.

Color intensity/foam observed was used to indicate the phytochemicals abundance.

There are so many unnecessary in-text citations. I have offered some suggestions in the attached draft file to allow for deletion of some that may not be useful.

Done

Please summarize your CONCLUSIONS.

Done

Journal Requirements:

1. Please complete all items on the Human Participants Research Checklist that are relevant for your submission, by following this link: http://journals.plos.org/plosone/s/file?id=dc11/PLOSOne_Human_Subjects_Research_Checklist.docx (Contact us at plosone@plos.org if you cannot access the document.) There may be overlap between the checklist items and other queries listed below; please address any duplicated queries both in your response email and on the checklist itself. Upload the completed Human Participants Research Checklist as file type “Other” when you re-submit your manuscript. This document is for internal journal use only and will not be published if your article is accepted. The requested information will help us to assess whether your submission complies with PLOS ONE’s policies and adheres to applicable reporting standards. Note that your manuscript may be rejected if you provide incomplete or inadequate responses to the checklist questions and that changing the ‘Section/Category’ of your article does not affect this requirement.

Reviewers' comments:

Reviewer's Responses to Questions

Comments to the Author

1. If the authors have adequately addressed your comments raised in a previous round of review and you feel that this manuscript is now acceptable for publication, you may indicate that here to bypass the “Comments to the Author” section, enter your conflict of interest statement in the “Confidential to Editor” section, and submit your "Accept" recommendation.

Reviewer #1: All comments have been addressed

Reviewer #5: All comments have been addressed

2. Is the manuscript technically sound, and do the data support the conclusions?

Reviewer #1: Yes

Reviewer #5: Partly

3. Has the statistical analysis been performed appropriately and rigorously?

Reviewer #1: Yes

Reviewer #5: Yes

4. Have the authors made all data underlying the findings in their manuscript fully available?

Reviewer #1: (No Response)

Reviewer #5: No

5. Is the manuscript presented in an intelligible fashion and written in standard English?

Reviewer #1: (No Response)

Reviewer #5: Yes

6. Review Comments to the Author

Reviewer #1: The authors addressed all comments. There are no new comments, and it can be accepted for publication in the journal.

Reviewer #5: Dear Author

Title: Antibacterial Activities of Aqueous and Ethanolic Extracts of Four Antioxidant Plants 2 Consumed by Western Lowland Gorilla (Gorilla gorilla gorilla).

Aim: The aim of this study was to determine the antioxidant and antibacterial activity of Ceiba pentandra, Myrianthus arboreus, Ficus ssp. and Milicia excelsa bark crude extracts (BCE), plants consumed by western lowland gorillas (Gorilla gorilla gorilla) and used in traditional medicine, and then to characterize their phytochemical compounds.

I have gone through the document.

These are my suggestions

In the introduction;

Please focus on the species of concern in relation to the study.

Its usage, values, usage and link everything to Gorillas

The desired medicinal properties should be aligned to the species of concern

I would like to believe that your study focuses on specific species. These should be addressed in the introduction.

Are the aspects mentioned from line 77 to 82 part of the study?

Once again try and bring focus to your study in relation to the aspects of interest as stated in the objectives

Ceiba pentandra, Myrianthus arboreus, Ficus subspecies (ssp.) and Milicia excelsa are the four species on which we focused our phytochemical investigations, for their antimicrobial and antioxidant activities, with a view to explaining the results obtained in a previous study on the antibiotic resistance of E. coli isolates from the feces of gorillas consuming these plants. Indeed, the results of this study showed that these animals were asymptomatic carriers of diarrheal E. coli strains multiresistant to antimicrobials used in human therapy.

In the light of these results, we conducted this second study to confirm or refute our hypothesis that the consumption of plants items, such as the nark, with antimicrobial and antioxidant activities could explain this asymptomatic character. This is one of the reasons why, in the introduction, we also focused on aspects of antimicrobial resistance, antioxidant and antimicrobial activities of medicinal plants, to highlight the interest of studying these four plants.

In addition, numerous other studies, which we have used as references, have already highlighted the importance of these plants in traditional medicine, due to their composition of bioactive phytochemicals.

Characterization as well as taxonomy, GPS coordinates of the species of interest is key

Unfortunately, during the course of this study, we did not have the necessary technical resources to characterize the active phytochemicals using techniques such as HPLC-MS.

As far as taxonomy is concerned, the studies referred to in the introduction have already highlighted the classification of the four plant species investigated in this study.

Concerning GPS coordinates, we have not received authorization to share or publish this kind of data, which is at the discretion of the government institutions responsible for managing these protected areas.

There is Table 1 which has the biggest list. There after we have table 2 with fewer species. There should a clear flow of information on the reasons. Otherwise you can just stick to the species of interest only and justify rather than having species that are not part of the study

Table 1 lists the various plants identified during a study on the phenologies of fruit-bearing plant species in Moukalaba-Doudou National Park, whose fruits and other organs are also consumed by the gorillas living there.

Table 2 contains only the four plants resulting from cross-searches in the oral and written literature (PubMed, Google scholar, Scopus, etc.), on their use in traditional medicine, but above all because they were the ones most cited during ethnopharmacological surveys carried out among the indigenous populations. Additional data from ethnobotanical and ethnopharmacological surveys are available in a supplementary file.

Discussion

State general observations

What caused those observations (positive and negative factors)

Compare with others, preferably recent

We have tried to take into account all your suggestions and recommendations, as well as those of the other reviewers, who explicitly instructed us to present the discussion in this form. We thank you for your understanding.

1. Mbehang Nguema, P.P., et al., High level of intrinsic phenotypic antimicrobial resistance in enterobacteria from terrestrial wildlife in Gabonese national parks. PLoS One, 2021. 16(10): p. e0257994.

2. Sigovini, M., E. Keppel, and D. Tagliapietra, Open Nomenclature in the biodiversity era. Methods in Ecology and Evolution, 2016. 7(10): p. 1217-1225.

---

## [Editor Report · Decision Letter 2]

27 Jun 2024

Antibacterial and Antioxidant Activities of Plants Consumed by Western Lowland Gorilla (Gorilla gorilla gorilla) in Gabon.

PONE-D-23-26722R2

Dear Dr. OYABA YINDA,

We’re pleased to inform you that your manuscript has been judged scientifically suitable for publication and will be formally accepted for publication once it meets all outstanding technical requirements.

Kind regards,

Timothy Omara, PhD

Academic Editor

PLOS ONE
---

## [Editor Report · Acceptance letter]

19 Aug 2024

PONE-D-23-26722R2 

PLOS ONE

Dear Dr. OYABA YINDA, 

I'm pleased to inform you that your manuscript has been deemed suitable for publication in PLOS ONE. Congratulations! Your manuscript is now being handed over to our production team.

Kind regards, 

on behalf of

Dr. Timothy Omara 

Academic Editor

PLOS ONE